# Unique integrated stress response sensors regulate cancer cell susceptibility when Hsp70 activity is compromised

Sara Sannino[1]*, Megan E Yates[2,3,4], Mark E Schurdak[5,6], Steffi Oesterreich[2,3,7], Adrian V Lee[2,3,7], Peter Wipf[8], Jeffrey L Brodsky[1]*

[1]Department of Biological Sciences, University of Pittsburgh, Pittsburgh, United States; [2]Women's Cancer Research Center, UPMC Hillman Cancer Center, Magee-Women Research Institute, Pittsburgh, United States; [3]Integrative Systems Biology Program, University of Pittsburgh, Pittsburgh, United States; [4]Medical Scientist Training Program, University of Pittsburgh School of Medicine, Pittsburgh, United States; [5]Department of Computational and Systems Biology, University of Pittsburgh, Pittsburgh, United States; [6]University of Pittsburgh Drug Discovery Institute, Pittsburgh, United States; [7]Department of Pharmacology and Chemical Biology, University of Pittsburgh School of Medicine, Pittsburgh, United States; [8]Department of Chemistry, University of Pittsburgh, Pittsburgh, United States

*For correspondence:
sannino.sara1986@gmail.com (SS);
jbrodsky@pitt.edu (JLB)

Competing interests: The authors declare that no competing interests exist.

**Abstract** Molecular chaperones, such as Hsp70, prevent proteotoxicity and maintain homeostasis. This is perhaps most evident in cancer cells, which overexpress Hsp70 and thrive even when harboring high levels of misfolded proteins. To define the response to proteotoxic challenges, we examined adaptive responses in breast cancer cells in the presence of an Hsp70 inhibitor. We discovered that the cells bin into distinct classes based on inhibitor sensitivity. Strikingly, the most resistant cells have higher autophagy levels, and autophagy was maximally activated only in resistant cells upon Hsp70 inhibition. In turn, resistance to compromised Hsp70 function required the integrated stress response transducer, GCN2, which is commonly associated with amino acid starvation. In contrast, sensitive cells succumbed to Hsp70 inhibition by activating PERK. These data reveal an unexpected route through which breast cancer cells adapt to proteotoxic insults and position GCN2 and autophagy as complementary mechanisms to ensure survival when proteostasis is compromised.

## Introduction

The many pathways that ensure the maintenance of protein homeostasis (proteostasis) must be tightly regulated to maintain cell function. Several proteostasis pathways are required to fold nascent or damaged proteins or mediate their degradation, which prevents the formation of toxic protein intermediates and aggregates that impair cellular homeostasis (*Balch et al., 2008*; *Mu et al., 2008*). The regulation and maintenance of cellular proteostasis is especially vital for cancer cells that thrive under hostile conditions, including anoxia, hypoglycemia, and oxidative stress (*Mei et al., 2013*; *Yan et al., 2015*). In addition, cancer cells grow uncontrollably, which requires high levels of protein biosynthesis, and are also aneuploid, producing orphaned proteins that should otherwise reside in multiprotein complexes (*Williams and Amon, 2009*). Each of these events disturb cellular proteostasis, thereby provoking cellular stress responses. As a result, tumor progression requires proteostasis adaptation and modulation of cellular stress responses (*Benbrook and Long, 2012*; *Cubillos-Ruiz et al., 2017*; *McConkey, 2017*; *Wang and Kaufman, 2014*).

One proteostatic mechanism to reduce proteotoxic stress is to slow translation. Indeed, reduced translation in cancer cells, which decreases the protein burden, allows for the selective production of proteins required for proliferation, tumor initiation, progression, and metastasis. This phenomenon commonly arises from phosphorylation of the alpha subunit of translation initiation factor 2 (eIF2$\alpha$) (*Blagden and Willis, 2011*; *Guo et al., 2017*; *Koromilas, 2015*; *Wek and Cavener, 2007*). In mammals, four kinases, the dsRNA-activated kinase (PKR), the heme-regulated inhibitor (HRI), the general control non-derepressible two factor (GCN2), and the PKR-like ER resident kinase (PERK), promote eIF2$\alpha$ phosphorylation (*Harding et al., 2000*; *Pakos-Zebrucka et al., 2016*; *Sidrauski et al., 2015*). The resulting integrated stress response (ISR) also facilitates cancer cell survival (*Darini et al., 2019*; *Sidrauski et al., 2015*; *Wek, 2018*), yet the role of each of the four ISR transducers in tumor initiation—or in maintenance of cellular homeostasis in most cells—is unclear. Moreover, how a single phosphorylation event integrates multiple stress stimuli and orchestrates different biological outcomes is incompletely understood.

Proteotoxic stress is especially problematic for secretory pathway function. Since cancer cells grow uncontrollably and require copious levels of growth factors, extracellular matrix components, and plasma membrane receptors and transporters, the unfolded protein response (UPR) mitigates the accumulation of toxic misfolded proteins in the endoplasmic reticulum (ER) in the cells (*Hetz et al., 2020*; *Ojha and Amaravadi, 2017*; *Ron and Walter, 2007*; *Yan et al., 2015*). The UPR is induced by three ER-localized transmembrane sensors, namely inositol-required enzyme 1 (IRE1), activating transcription factor 6 (ATF6), and PERK, which as noted above is also a component of the ISR (*Bi et al., 2005*; *Walter and Ron, 2011*). Among other events, the UPR favors the production of pro-folding molecular chaperones, augments protein secretion, and increases ER-associated degradation as well as lipid synthesis, which expands the ER. Together, the ISR and the UPR reestablish proteostasis or can alternatively initiate a cell death pathway if stress cannot be rectified (*Bi et al., 2005*; *Nam and Jeon, 2019*; *Oyadomari and Mori, 2004*; *Walter and Ron, 2011*). How cancer cells and most other cells avoid ISR/UPR-activated apoptosis is also poorly understood (*Cerezo et al., 2016*; *Cubillos-Ruiz et al., 2017*; *Lin et al., 2019*).

To mitigate misfolded protein toxicity, the activity of the ubiquitin-proteasome and autophagy pathways also rise upon UPR/ISR induction in cancer cells (*Clarke et al., 2011*; *Kim et al., 2015*; *Schönthal, 2012*). These protective pathways additionally favor amino acid recycling and cancer cell survival (*Amaravadi et al., 2016*; *B'chir et al., 2013*; *Levine and Kroemer, 2008*; *Ma et al., 2014*; *Mizushima et al., 2008*). Based on the vital roles played by the UPR, ISR, and protein degradation pathways—and their known links to cell proliferation, apoptosis, chemotherapeutic resistance, and metastasis—inhibitors of these pathways represent new therapeutic targets (*Deshaies, 2014*; *Grandjean and Wiseman, 2020*; *Hazari et al., 2016*; *Lazova et al., 2012*; *Sharifi et al., 2016*) Molecular chaperones, via their ability to augment protein folding and protect cells from proteotoxic damage, are also induced in cancer cells. In fact, molecular chaperones, such as Hsp70 and Hsp90, are predictors of poor prognoses in cancer patients and have emerged as new therapeutic targets (*Calderwood and Gong, 2016*; *Joshi et al., 2018*; *Powers et al., 2009*; *Sannino and Brodsky, 2017*; *Whitesell and Lindquist, 2005*).

Through the application of the Hsp70 inhibitor, MAL3-101, we previously reported that Hsp70 plays a crucial role in the survival of rhabdomyosarcoma, a childhood cancer that is largely refractory to conventional therapies (*Sabnis et al., 2016*; *Sannino et al., 2018*). MAL3-101 binds a unique site in Hsp70 where it blocks allosteric activation by the Hsp40 co-chaperone. The compound, a pyrimidinone peptoid, primarily targets cytosolic Hsp70 with minor effects on the activity of the ER resident Hsp70 chaperone, BiP, and lacks toxicity in non-transformed cells and rodents (*Adam et al., 2014*; *Fewell et al., 2004*; *Patham et al., 2009*; *Sabnis et al., 2016*; *Singh et al., 2020*; *Wisén et al., 2010*). MAL3-101-mediated Hsp70 inhibition led to the accumulation of p-eIF2$\alpha$ and CHOP in rhabdomyosarcoma cell lines and in a mouse model, thereby initiating an apoptotic response (*Sabnis et al., 2016*). Interestingly, components of the autophagy pathway and autophagic flux appeared to be higher in Hsp70 inhibitor resistant cells that were isolated after drug escalation (*Sannino et al., 2018*). However, the molecular mechanisms that led to Hsp70 inhibitor resistance and thereby rewired the rhabdomyosarcoma proteostasis pathways were unclear. A deeper understanding of how drug resistance arises when cancer cells are challenged with proteostasis inhibitors is critical for enabling future clinical applications.

In contrast to rhabdomyosarcoma, which is relatively rare, breast cancer represents one of the most common causes of death worldwide (*Bray et al., 2018*; *Kohler et al., 2015*). Breast cancer is a heterogeneous group of tumors subdivided into luminal, HER2, and Triple Negative Breast Cancer (TNBC) types depending on the expression of the estrogen receptor, progesterone receptor, and epidermal growth factor receptor 2 (HER2) (*Bareche et al., 2018*; *Cancer Genome Atlas Network, 2012*; *Perou et al., 2000*). We now report that breast cancer cells bin into distinct groups when challenged with the specific Hsp70 inhibitor, MAL3-101. Although both resistant and sensitive breast cancer cell lines accumulated p-eIF2α when Hsp70 activity was thwarted, the resistant cells exclusively required the autophagy pathway to survive when challenged with MAL3-101. We then discovered that the ISR sensor, GCN2, was also essential for drug resistance. In contrast, the dual UPR/ISR sensor, PERK, initiated cell death in inhibitor-sensitive cells. These findings delineate distinct roles of ISR inducers in cell survival and highlight the interplay between the ISR/UPR, Hsp70, and protective protein degradation pathways.

## Results

### Breast cancer cells exhibit a range of sensitivities to Hsp70 inhibition

Elevated Hsp70 is associated with resistance to chemotherapy in several cancers (*Brondani Da Rocha et al., 2004*; *Nanbu et al., 1998*; *Vargas-Roig et al., 1998*), and Hsp70 overexpression correlates with metastasis in breast cancer murine models and in patients (*Powers et al., 2010*; *Sun et al., 2008*). To determine if Hsp70 activity is required for breast cancer cell survival, we performed a pilot study in which sensitivity to the Hsp70 inhibitor, MAL3-101 (*Fewell et al., 2004*), was measured in a panel of 14 breast cancer lines (*Figure 1A* and *Table 1*). MAL3-101 targets an allosteric site in the ATP-binding domain of Hsp70, specifically inhibits the ability of Hsp40 co-chaperones to activate Hsp70 ATPase activity, and has demonstrated *in vitro* and *in vivo* efficacy in Merkel Cell carcinoma and multiple myeloma as well as rhabdomyosarcoma cell lines (*Adam et al., 2014*; *Braunstein et al., 2011*; *Sabnis et al., 2016*; *Sannino et al., 2018*; *Wisén et al., 2010*). Among the luminal, HER2, and TNBC breast cancer subtypes, we were able to bin each line into sensitive, intermediate, and resistant groups (*Figure 1A* and *Table 1*). Breast cancer cells with an $IC_{50} \leq 3.5$ µM were considered MAL3-101 sensitive, whereas cells with an $IC_{50} \geq 30$ µM were considered resistant. $IC_{50}$s of breast cancer cells with an intermediate sensitivity were between 4 µM and 14 µM (*Table 1*). Interestingly, among identical breast cancer subtypes, breast cancer cells could be sensitive or insensitive to the Hsp70 inhibitor, indicating that other features dictate sensitivity to increased proteotoxic stress.

To confirm these results, we measured the levels of apoptosis markers in two representative MAL3-101-sensitive lines (*Figure 1B,C*; MDA MB 231 and MCF7, indicated in blue text) and resistant lines (MDA MB 453 and MDA MB 361, indicated in black text). In each group, one cell line represents a luminal cancer cell type (MCF7 and MDA MB 361) and one is categorized as a TNBC cell line (MDA MB 231 and MDA MB 453). Induction of apoptosis in the MAL3-101-sensitive line, MDA MB 231, was evident by the accumulation of cleaved caspase-3, caspase-7, and caspase-8. Because MCF7 cells are caspase-3 deficient (*Kagawa et al., 2001*; *Turner et al., 2003*), the cleavage of only caspase-7 was detected after Hsp70 inhibition (*Figure 1B,C*). In contrast, apoptotic markers were absent in the MDA MB 453 and MDA MB 361-resistant lines. MAL3-101 treatment also increased Hsp70 mRNA abundance (*Figure 1—figure supplement 1A*), and a variable but general increase in Hsp70 protein levels was apparent (*Figure 1—figure supplement 1B*) that failed to correlate with Hsp70 inhibitor sensitivity and resistance. The muted levels of Hsp70 protein induction compared to the massive increase (~20–40-fold) in gene expression are consistent with previous data in which a heat shock trigger was administered to cells in culture and the effects on Hsp70 protein and mRNA levels were examined (*Petersen and Lindquist, 1989*; *Theodorakis and Morimoto, 1987*). A general increase in Hsp70 mRNA is also consistent with MAL3-101-dependent inhibition of Hsp70, which compromises proteostasis (*Sabnis et al., 2016*; *Sannino et al., 2018*). In addition, when the steady-state abundance of cytoplasmic Hsp70 and the constitutively expressed isoform, Hsc70, were measured (*Figure 1—figure supplement 1C*), no correlation between protein levels and MAL3-101 sensitivity was apparent. These data strongly suggest that sensitivity/resistance arise from an independent phenomenon.

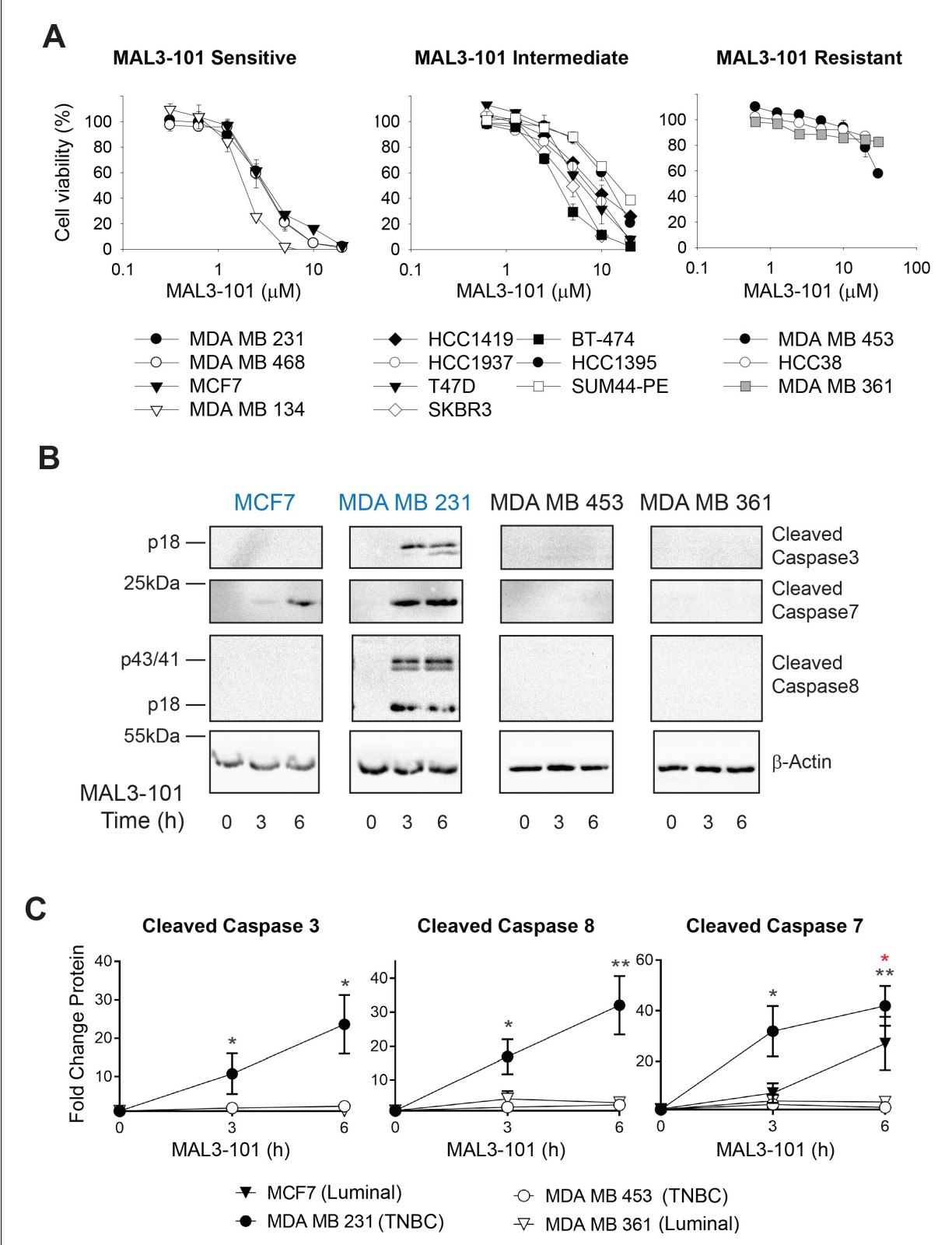

**Figure 1.** Breast cancer cell lines exhibit unique sensitivities to an Hsp70 inhibitor, MAL3-101. (**A**) HER2 (diamonds), TNBC (circles), and luminal (squares and triangles) breast cancer lines were seeded into 96-well plates and treated with increasing doses of MAL3-101 for 72 hr. Viability is expressed as the average of three or more independent experiments, ± SEM. (**B**) MAL3-101-sensitive (MCF7 and MDA MB 231, denoted in blue) and resistant (MDA MB 453 and MDA MB 361, denoted in black) cells were treated with 12 μM MAL3-101 for the indicated times, and lysates were prepared and

*Figure 1 continued on next page*

*Figure 1 continued*

immunoblotted for cleaved caspase-3, caspase-7, and caspase-8. β-actin serves as a loading control. (C) The corresponding fold-increase of the indicated apoptotic markers relative to the DMSO control are plotted, ± SEM (n≥3 for cleaved caspase-3, n=3 for cleaved caspase-7, and n≥4 for cleaved caspase-8). Black asterisks correspond to statistical significance between MDA MB 231 cells (closed circle) and MDA MB 453 and MDA MB 361 (open circle and triangle, respectively), and the red asterisk represents statistical significance between MCF7 (closed triangle) and MDA MB 453 and MDA MB 361 (open circle and triangle) cells; * denotes p<0.05, ** denotes p<0.005.

The online version of this article includes the following source data and figure supplement(s) for figure 1:

**Source data 1.** Source data for cell viability assay and apoptotic marker accumulation in *Figure 1*.
**Figure supplement 1.** MAL3-101 sensitivity is independent of Hsp70 levels or induction.
**Figure supplement 1—source data 1.** Raw data for Hsp70 mRNA and protein abundance.

Hsp70 is a critical component of the Hsp90 'chaperome' and facilitates the folding of oncogenic proteins such as BRAF, HER2, and AKT (*Basso et al., 2002*; *Joshi et al., 2018*; *Sannino and Brodsky, 2017*; *Schopf et al., 2017*; *Whitesell and Lindquist, 2005*; *Xu et al., 2005*). To examine whether Hsp90 and Hsp70 inhibitor sensitivities correlate, we applied the Hsp90 inhibitor 17-AAG (17-allylamino-17-demethoxy-geldanamycin, also known as tanespimycin) (*Doddareddy et al., 2011*; *Schulte and Neckers, 1998*; *Taldone et al., 2009*) to each cell line. In accordance with previous reports (*Citri et al., 2004*; *Patel et al., 2013*; *Xu et al., 2005*), HER2-expressing breast cancer cells were more sensitive to 17-AAG (*Figure 1—figure supplement 1C* and *Table 1*), but again there was no correlation with MAL3-101 sensitivity/resistance. Thus, the compounds appear to selectively affect different downstream effectors and compensatory mechanisms.

## Enhanced protein degradation pathways compensate for impaired proteostasis

Hsp70 plays an essential role in cellular protein folding, degradation, transport, modification, and assembly (*Balchin et al., 2016*; *Evans et al., 2010*; *Goloudina et al., 2012*; *Rosenzweig et al., 2019*). Thus, reducing Hsp70 activity favors the accumulation of unfolded proteins. However, induction of compensatory proteostatic pathways, such as proteasome-dependent degradation and autophagy, might temper sensitivity to Hsp70 inhibitors. We previously demonstrated that rhabdomyosarcoma cells are hypersensitive to Hsp70 inhibition and acquired MAL3-101 resistance as a

**Table 1.** Breast cancer cells exhibit a range of sensitivities to MAL3-101, a specific Hsp70 inhibitor.
The indicated breast cancer lines were seeded into 96-well plates and treated with increasing doses of the indicated compounds for 72 hr. Viability was measured using the CellTiter-Glo assay. $IC_{50}$ values were generate using a sigmoidal nonlinear regression with SigmaPlot 11.0. ND stands for an undetermined value. MAL3-101 sensitivities of Hsp70 inhibitor resistant cells are in bold.

| Breast cancer subtype | Cell line | MAL3-101 | CQ | Bafilomycin | Bortezomib | CB-5083 | 17-AAG | VER155008 |
|---|---|---|---|---|---|---|---|---|
| TNBC | MDA MB 231 | 3.3 µM | 40 µM | 2.0 nM | 5.9 nM | 0.6 µM | 1.3 µM | 18.3 µM |
| | MDA MB 468 | 3.1 µM | 30 µM | 2.0 nM | 5.2 nM | 0.8 µM | 1.5 µM | >100 µM |
| | HCC1937 | 7.5 µM | 42 µM | 5.9 nM | 5.0 nM | 1.0 µM | 5.6 nM | ND |
| | HCC1395 | 11.6 µM | 16 µM | 3.0 nM | 9.2 nM | 0.3 µM | ND | ND |
| | **HCC38** | **>30 µM** | 60 µM | 2.7 nM | 6.0 nM | 0.5 µM | 17.6 nM | >100 µM |
| | **MDA MB 453** | **>30 µM** | 23 µM | 2.0 nM | 4.3 nM | 1.1 µM | 13.5 nM | >100 µM |
| Luminal | MDA MB 134 | 1.9 µM | 18 µM | 8.2 nM | 3.7 nM | ND | ND | ND |
| | MCF7 | 3.0 µM | 10 µM | 1.0 nM | 40.0 nM | 0.5 µM | 63 nM | >100 µM |
| | T47D | 6.6 µM | 18 µM | 2.7 nM | 9.8 nM | ND | 1.3 µM | >100 µM |
| | BT-474 | 4.0 µM | 12 µM | 2.5 nM | 16.3 nM | ND | 15.8 nM | >100 µM |
| | SUM44-PE | 13.5 µM | 42 µM | 1.5 nM | 16.0 nM | ND | ND | ND |
| | **MDA MB 361** | **>30 µM** | 13 µM | 1.7 nM | 13.0 nM | 1.0 µM | 25.8 nM | >100 µM |
| HER2 | SKBR3 | 4.5 µM | 28 µM | 2.0 nM | 4.4 nM | 0.6 µM | 18.5 nM | >100 µM |
| | HCC1419 | 8.2 µM | 34 µM | 1.5 nM | 7.8 nM | 0.8 µM | 54.5 nM | 100 µM |

result of increased protein degradation pathway activity (*Sannino et al., 2018*). To define which compensatory pathway(s) might be altered and thus restrict breast cancer cell death in the presence of MAL3-101, we first asked if proteasome-dependant protein turnover was higher in MAL3-101 resistant lines. To this end, we measured the levels of ubiquitinated proteins in sensitive (MDA MB 231 and MCF7, in blue), intermediate (BT-474, in purple), and resistant (MDA MB 453 and MDA MB 361, in black) cells in the presence or absence of the proteasome inhibitor bortezomib (also known as Velcade) (*Chen et al., 2011*). As expected, bortezomib-mediated proteasome inhibition increased polyubiquitinated protein accumulation in all cell lines, but the level of total ubiquitinated proteins was greater in MAL3-101-resistant cells compared to cell lines with increased MAL3-101 sensitivity (*Figure 2A*, compare lanes 2 and 4 with lanes 8 and 10). These data suggest that resistant lines rely to a greater extent on the removal of misfolded proteins to survive.

Next, to determine whether proteasome activity might be higher in resistant cells, we prepared lysates and examined proteasome activity by measuring the MG 132-dependent signal produced from the fluorogenic substrate, Suc-LLVY-7-amino-4-methylcoumarin (*Stein et al., 1996*). As shown in *Figure 2—figure supplement 1A*, no statistically significant differences in proteasome activity were detected between the sensitive and resistant lines. These data indicate that even though MAL3-101-resistant cells rely more heavily on protein degradation pathways to mitigate the accumulation of misfolded proteins (*Figure 2A*), overall proteasome activity is unchanged in Hsp70 inhibitor sensitive and resistant cells under non-stress conditions. The increased accumulation of polyubiquitinated protein in MAL3-101-resistant cells might be due to a differential, compensatory involvement of the ubiquitination machinery between MAL3-101 sensitive and resistant lines upon proteasome inhibition, perhaps even reflecting altered E3 activity. Interestingly, however, the growth of the MAL3-101 sensitive and resistant lines displayed a range of sensitivities to the proteasome inhibitor bortezomib (*Figure 2B* and *Table 1*), but sensitivity still failed to correlate with Hsp70-inhibitor sensitivity. These results suggest that inhibition of the proteasome and Hsp70 kill breast cancer cells through independent mechanisms.

Another cellular quality control pathway that modulates proteostasis is the endoplasmic reticulum associated degradation pathway (ERAD). ERAD substrates are recognized by chaperones, including Hsp70, ubiquitinated at the ER membrane, retrotranslocated to the cytoplasm in a p97-dependent manner, and delivered to the proteasome (*Bagola et al., 2011*; *Needham and Brodsky, 2013*; *Smith et al., 2011*; *Tsai and Weissman, 2011*). It was previously reported that p97 expression is higher in HER2 positive breast cancer cells than other cells, suggesting that ERAD contributes to HER2 breast cancer cell survival (*Singh et al., 2015*). Therefore, we tested ERAD pathway dependence in resistant and sensitive breast cancer cells by applying a clinical drug candidate, CB-5083, which targets the p97 ATPase domain (*Zhou et al., 2015*). As shown in *Figure 2—figure supplement 1B*, minimal differences were detected in CB-5083 sensitivity between the lines (*Table 1*). Moreover, in contrast to previous work (*Singh et al., 2015*), the sensitivities of the examined HER2 positive and HER2 negative breast cancer cell lines were comparable (*Table 1*).

In contrast to ERAD, the autophagy pathway destroys not only misfolded and long-lived proteins, but also protein aggregates and damaged organelles (*Amaravadi et al., 2016*; *B'chir et al., 2013*; *Cavalli and Cenci, 2020*; *Mizushima, 2017*; *Towers and Thorburn, 2016*). During this process, cytoplasmic proteins are captured in double-membrane autophagosomes, and upon fusion with lysosomes the vesicle contents are degraded in autophagolysosomes (also termed autolysosomes). This pathway can be stimulated by cellular stressors, including nutrient or growth factor deprivation, the UPR, hypoxia, reactive oxygen species, DNA damage, drug treatments, or after pathogen invasion (*He and Klionsky, 2009*; *Kroemer et al., 2010*; *Miracco et al., 2010*; *Rubinsztein et al., 2007*; *Wojcik, 2013*). Moreover, autophagy can serve as a back-up for the ubiquitin-proteasome pathway (*Fan et al., 2018*; *Kocaturk and Gozuacik, 2018*; *Wojcik, 2013*) as well as the ERAD pathway (*Ishida et al., 2009*; *Kruse et al., 2006*). Therefore, we first asked if MAL3-101 sensitive and resistant breast cancer cells exhibited unique sensitivities to chloroquine (CQ) and bafilomycin, which block the final step in the autophagy pathway and impair the degradation of autophagosomal contents (*Ahlberg et al., 1985*; *Klionsky et al., 2016*; *Manic et al., 2014*; *Mizushima et al., 2011*). However, Hsp70 inhibitor resistance failed to correlate with increased CQ and bafilomycin sensitivity (*Figure 2C* and *Figure 2—figure supplement 1C*; also see *Table 1*). We next measured the abundance of key autophagy markers, LC3B and ATG5–ATG12 (*Kabeya et al., 2000*; *Pyo et al., 2005*). Immunoblot analysis showed greater amounts of ATG5–ATG12 in the resistant MDA MB 453 and

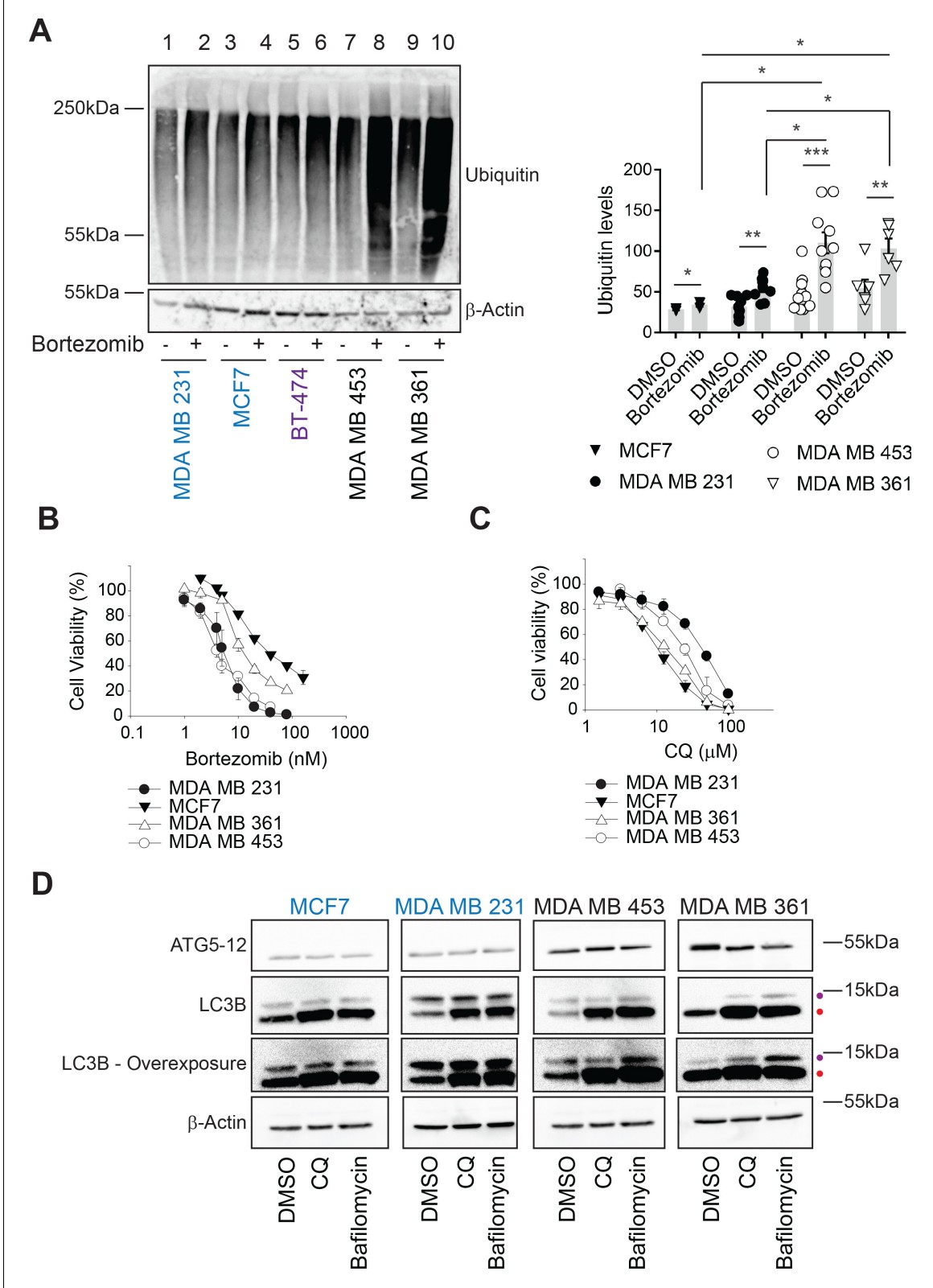

**Figure 2.** Ubiquitinated proteins are lower in MAL3-101-resistant breast cancer cells. (**A**) MAL3-101 sensitive (in blue), intermediate (in purple), and resistant (in black) lines were treated with DMSO or bortezomib for 4 hr followed by immunoblot analysis to detect total levels of ubiquitinated proteins in the cell. β-actin serves as a loading control. Data for MAL3-101 sensitive and resistant lines (raw counts from imaging) are plotted in the graph to the right, ± SEM (n=8 for MDA MB 231, n=3 for MCF7, n=10 for MDA MB 453, and n=6 for MDA B 361). Normalized ubiquitin levels relative to β-actin

*Figure 2 continued on next page*

*Figure 2 continued*

signals are plotted in the graph. There was a ~2-fold increase in ubiquitinated protein levels in MAL3-101 resistant cells (open symbols) versus the MAL3-101-sensitive lines (closed symbols) in the presence of bortezomib. * denotes p<0.05, ** denotes p<0.005, and *** denotes p<0.0005. (B–C) MAL3-101-sensitive lines (closed symbols) and resistant cells (open symbols) were seeded into 96-well plates and treated with increasing doses of (B) bortezomib, a proteasome inhibitor, or (C) chloroquine (CQ), an autophagy inhibitor, for 72 hr. Cell viability data represent the average of three or more independent experiments, ± SEM. (D) The levels of autophagy related proteins in MAL3-101 sensitive (in blue) and resistant (in black) lines were analyzed by immunoblotting in presence or absence of CQ or bafilomycin for 6 hr. Purple and red dots respectively indicate the soluble (LC3BI) and the autophagosome-associated isoform (LC3BII) of LC3B. An overexposed image is also shown to better visualize the LC3BI and LC3BII isoforms in the cell lines.

The online version of this article includes the following source data and figure supplement(s) for figure 2:

**Source data 1.** Source data for *Figure 2*.
**Figure supplement 1.** Autophagy pathway activity is higher in MAL3-101-resistant cells.
**Figure supplement 1—source data 1.** Raw data for *Figure 2—figure supplement 1*.

MDA MB 361 lines compared to the MDA MB 231 and MCF7-sensitive cell lines (*Figure 2D*), suggesting that the elongation phase of autophagy was induced in resistant cells (*Mizushima et al., 2011*; *Strømhaug et al., 2004*). Consistent with this hypothesis, lipidated LC3B (LC3BII), which is appended to autophagosomal membranes, accumulated to a greater extent in the resistant (MDA MB 453 and MDA MB 361) cells when autophagy was inhibited by CQ or bafilomycin (*Figure 2D*, indicated by a red dot, and quantified in *Figure 2—figure supplement 1D*). Moreover, the conversion of LC3BI (the soluble LC3B isoform) into LC3BII was enhanced in resistant cells (*Figure 2D* and *Figure 2—figure supplement 1E*), as evidenced by the increased LC3BII:LC3BI ratio (*Barth et al., 2010*; *McLeland et al., 2011*; *Zhang et al., 2016*). The magnitude of this effect is consistent with other studies demonstrating autophagy induction (*Klionsky et al., 2016*; *Li et al., 2018*; *Rouschop et al., 2010*; *Sannino et al., 2018*).

Cancer cells induce autophagy to survive nutrient deprivation (*Amaravadi et al., 2016*; *Klionsky, 2005*; *Mizushima et al., 2008*). Thus, we asked whether autophagy pathway activation was also higher in resistant cells upon starvation. As shown in *Figure 2—figure supplement 1F*, LC3BII accumulated to a greater extent when CQ was added to resistant cells under short-term starvation in nutrient-depleted (EBSS) medium compared to sensitive lines. Together, these data indicate that autophagic flux is magnified in the inhibitor resistant (MDA MB 453 and MDA MB 361) breast cancer cells.

## Autophagy inhibition re-sensitizes resistant cells to Hsp70 inhibitor

The data in the previous section suggest that misfolded proteins are directed to both the ubiquitin-proteasome and autophagy pathways, but only autophagy favors the survival of Hsp70 inhibitor resistant cells in the presence of MAL3-101. Therefore, we asked if modest inhibition of the autophagy and ubiquitin-proteasome pathways increased Hsp70 inhibitor sensitivity. First, MAL3-101 sensitivity in the presence of sub-maximal concentrations of either CQ or bortezomib in MAL3-101 sensitive and resistant lines was examined (see *Table 2* for CQ and bortezomib doses). As shown in *Figure 3A*, little effect was seen in the sensitive (MDA MB 231 and MCF7) cells, but CQ more prominently increased MAL3-101 sensitivity in the resistant (MDA MB 453 and MDA MB 361) line. A similar phenomenon was observed when MAL3-101 was administered in presence of a sub-maximal dose of bafilomycin (*Figure 3—figure supplement 1A* and *Tables 2* and *3*). Second, we found that bortezomib modestly increased the sensitivity of both a sensitive (MDA MB 231) and a resistant (MDA MB 453) cell line, but to a significantly lesser extent compared to CQ (*Figure 3B*). No increase in sensitivity to MAL3-101 was apparent in the presence of bortezomib in the other lines (*Figure 3B*). Consequently, a combination of enhanced proteotoxic stress, via the addition of MAL3-101, and sub-optimal levels of autophagy can re-sensitize otherwise resistant cells to an Hsp70 inhibitor.

To support this conclusion, we silenced ATG5 in MAL3-101-resistant cells with two different siRNA oligonucleotides. ATG5 facilitates the autophagy elongation step and, together with ATG12 and ATG16L, links LC3BII to the growing autophagosomal membrane (*Acevo-Rodríguez et al., 2020*; *Cuervo, 2004*; *Mizushima et al., 2011*). While MAL3-101 sensitivities were unaffected by the transfection of a control siRNA, ATG5 knockdown decreased ATG5-12 abundance by ~70% and again re-sensitized the resistant cells to MAL3-101 (*Figure 3—figure supplement 1B and C*). As

**Table 2.** The cell numbers and autophagy or proteasome inhibitor concentrations used for the cell viability assay in combination with increasing doses of MAL3-101 are shown.

The concentrations of bortezomib, CQ, and bafilomycin to induce no greater than 30% of cell death in each line after 72 hr treatment are shown. ND stands for undetermined value.

| Breast cancer subtype | Cell line | Number of cells/well | Everolimus | CQ | Bafilomycin | Bortezomib |
|---|---|---|---|---|---|---|
| TNBC | MDA MB 231 | 3000 | 2.0 µM | 14 µM | 0.8 nM | 4.0 nM |
| | MDA MB 468 | 2500 | ND | 14 µM | 0.6 nM | 2.7 nM |
| | HCC1937 | 2000 | ND | 21 µM | 2.2 nM | 2.0 nM |
| | HCC1395 | 3000 | ND | 5.0 µM | 1.6 nM | 3.5 nM |
| | HCC38 | 3000 | ND | 33 µM | 1.0 nM | 3.0 nM |
| | MDA MB 453 | 3000 | 2.0 µM | 12 µM | 1.0 nM | 2.5 nM |
| Luminal | MDA MB 134 | 5000 | ND | 15 µM | 6.6 nM | 2.5 nM |
| | MCF7 | 1000 | ND | 7 µM | 0.5 nM | 13.0 nM |
| | T47D | 2000 | ND | 9.5 µM | 1.3 nM | ND |
| | BT-474 | 3000 | ND | 6.0 µM | 1.4 nM | 10.2 nM |
| | SUM44-PE | 8000 | ND | 25 µM | ND | 7.0 nM |
| | MDA MB 361 | 4500 | ND | 8.0 µM | 1.0 nM | 6.5 nM |
| HER2 | SKBR3 | 2500 | ND | 13 µM | 1.0 nM | 2.5 nM |
| | HCC1419 | 4000 | ND | 12 µM | 0.8 nM | 2.5 nM |

noted above, no significant changes in MAL3-101 sensitivity were detected in the MDA MB 231 sensitive cell line and only a slight increase in MCF7 cell sensitivity was apparent upon ATG5 knockdown.

## Hsp70 inhibition triggers autophagy pathway induction in resistant breast cancer cells

Based on a potential compensatory role played by the autophagy pathway in cells with magnified proteotoxic stress, we investigated if autophagy is also acutely activated when Hsp70 function is muted. When the abundance of select autophagy-related genes and the gene encoding LC3B (MAP1LC3B) were measured 6 hr after MAL3-101 was added, the expression levels of ATG3, ATG5, ATG12, and MAP1LC3B were increased to a significantly higher extent in the resistant (MDA MB 453 and MDA MB 361) cell lines compared to the sensitive (MDA MB 231 and MCF7) cells (*Figure 4A*). Of note, the magnitude of the effect is in accordance with the increased levels of autophagy related genes detected in other studies (*B'chir et al., 2013*; *Füllgrabe et al., 2016*; *Sannino et al., 2018*). RNA collected from cells was also used to analyze the time-dependent, relative amplitude of the transcriptional response in the MAL3-101 sensitive and resistant cells 3 and 6 hr after Hsp70 was inhibited. Quantitative PCR analysis showed that each message rose over time to a greater degree in MAL3-101 resistant cells (*Figure 4B*). That autophagy was induced primarily in the resistant cells was confirmed by quantifying the abundance of LC3BII and the conversion rate of LC3BI in LC3BII when autophagy was inhibited with CQ (*Figure 4C*). In line with gene expression data, autophagy was also elevated more prominently in the resistant cells (MDA MB 453 and MDA MB 361) after Hsp70 activity was compromised with MAL3-101 for 6 hr (*Figure 4C and D*, see relative 'CQ' levels).

To better characterize the effect of Hsp70 inhibition on autophagic flux, we engineered a sensitive (MDA MB 231) and resistant (MDA MB 453) cell line to stably express an mRFP-GFP-tagged LC3B autophagy reporter (also known as tfLC3) (*Kimura et al., 2007*). Dual color analysis with this reporter provides a direct measure of autophagic flux (*Figure 5A*). In brief, when mRFP-GFP-LC3 decorated autophagosomes fuse with lysosomes, the lower intravesicular pH quenches GFP, permitting visualization of red fluorescence (via the mRFP fluorophore) and a rise in RFP+GFP- puncta. An increase in the number of RFP+GFP- puncta therefore indicates more robust delivery of

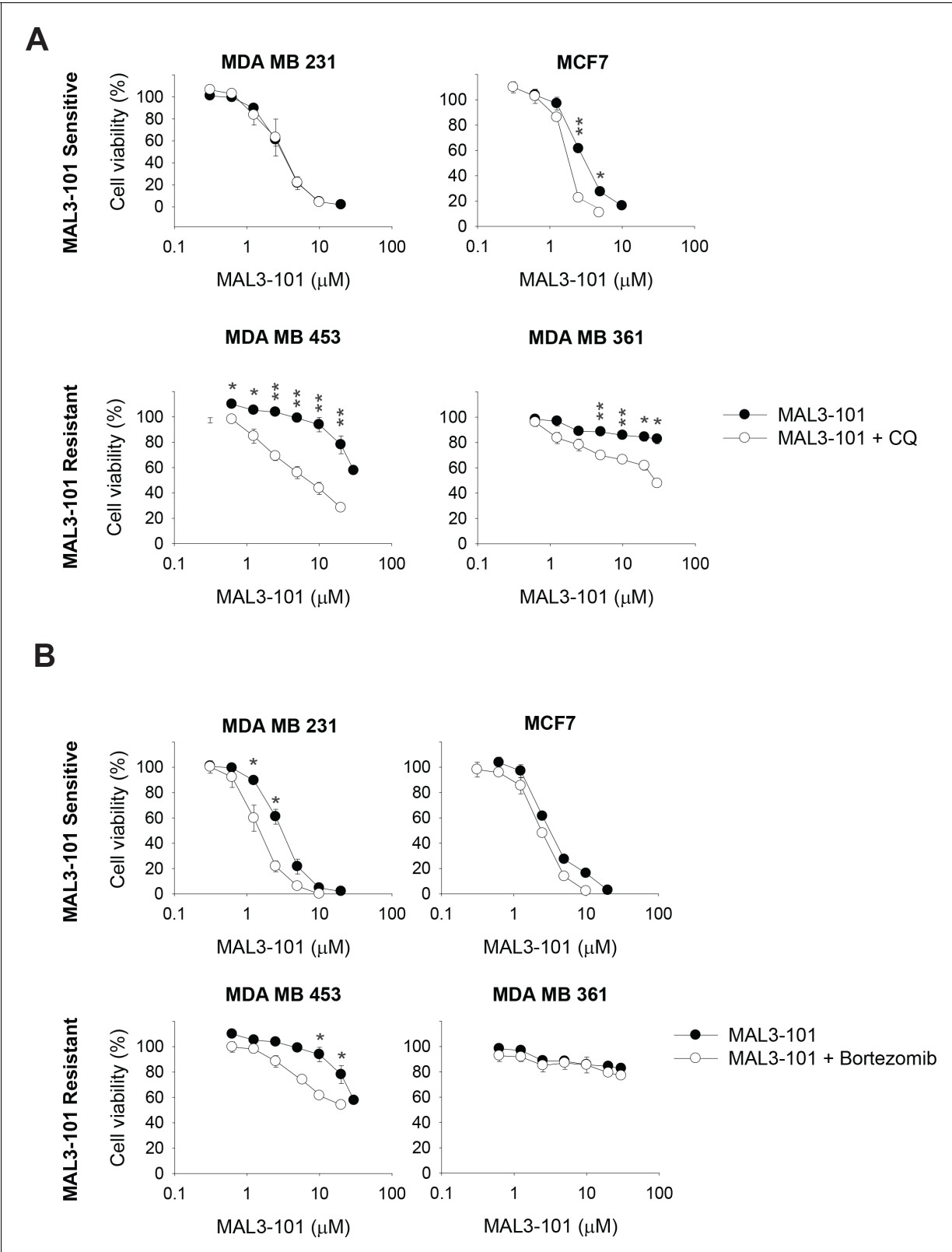

**Figure 3.** The autophagy pathway contributes to MAL3-101 resistance. MAL3-101 sensitive and resistant lines (closed and open symbols, respectively) were treated with increasing doses of MAL3-101 in the presence or absence of a sub-lethal dose of (**A**) CQ or (**B**) bortezomib (see *Table 2* and *Table 3*). Cell viability was detected at 72 hr, and data represent the means of three independent experiments, ± SEM. * denotes p<0.05, ** denotes p<0.005.

*Figure 3 continued on next page*

*Figure 3 continued*

The online version of this article includes the following source data and figure supplement(s) for figure 3:

**Source data 1.** Source data for cell viability assays in *Figure 3*.
**Figure supplement 1.** Autophagy inhibition re-sensitizes resistant cells to MAL3-101.
**Figure supplement 1—source data 1.** Source data for *Figure 3—figure supplement 1*.

autophagosomes to lysosomes, and consequently an increase in autophagolysosome formation (*Klionsky et al., 2016*; *Zhang et al., 2016*).

As shown in *Figure 5B* (and quantified in *Figure 5C*), Hsp70 inhibition favored the accumulation of RFP+GFP- puncta, which are representative of increased autophagolysosome formation in resistant cells (MDA MB 453), but not in the MAL3-101 sensitive line (MDA MB 231). These data confirm that Hsp70 inhibition induces autophagy only in MAL3-101-resistant cells. In contrast, the percentage of RFP+GFP- puncta in both MAL3-101 sensitive and resistant lines increased upon starvation in EBSS media (*Figure 5B and C*), indicating that autophagy flux remains competent in sensitive cells (*Figure 5B and C*). Because the percentage of RFP+GFP- puncta per cell was comparable after MAL3-101 and EBSS treatment in the resistant (MDA MB 453) cells, the magnitude of the effect of MAL3-101 is akin to that triggered by nutrient depletion. We conclude that Hsp70 inhibition acutely induces autophagy in the MAL3-101-resistant line to compensate for proteotoxic stress.

## Combined inhibition of autophagy and Hsp70 leads to apoptotic cell death

We next investigated whether CQ and MAL3-101 not only prevented cell growth but also triggered apoptosis in resistant cells. MAL3-101 treatment alone was unable to induce apoptosis in the MAL3-101-resistant line (MDA MB 453). In contrast, cleaved caspase-3, caspase-7, and caspase-8 along with the pro-apoptotic protein, CHOP, accumulated when CQ was administrated for 2 hr to MAL3-101-treated cells (*Figure 6A* and *Figure 6—figure supplement 1A*). As anticipated by monitoring cell growth (*Figure 3*), induction of the apoptotic pathway in the MDA MB 231 (sensitive) cell line

**Table 3.** Breast cancer cells exhibit a range of sensitivities to MAL3-101 in the presence of either autophagy or proteasome inhibitors.

Cells were seeded into 96-well plates and treated with increasing doses of MAL3-101 in the presence or absence of subcritical doses of bortezomib (proteasome inhibitor), or CQ or bafilomycin (autophagy inhibitors) for 72 hr. Viability was measured using the CellTiter-Glo assay. $IC_{50}$ values were generate using a sigmoidal nonlinear regression with SigmaPlot 11.0. ND stands for undetermined value. MAL3-101 resistant cells are highlighted in yellow.

| Breast cancer subtype | Cell line | MAL3-101 | MAL3-101 + CQ | MAL3-101 + Bafilomycin | MAL3-101 + Bortezomib | MAL3-101+ Everolimus |
|---|---|---|---|---|---|---|
| TNBC | MDA MB 231 | 3.3 µM | 3.3 µM | 3.4 µM | 1.5 µM | 5.1 µM |
| | MDA MB 468 | 3.1 µM | 3.0 µM | 3.2 µM | 1.8 µM | ND |
| | HCC1937 | 7.5 µM | 5.1 µM | 6.4 µM | 7.1 µM | ND |
| | HCC1395 | 11.6 µM | 5.8 µM | 3.9 µM | 10.6 µM | ND |
| | HCC38 | >30 µM | 17.0 µM | 20.0 µM | >30 µM | ND |
| | MDA MB 453 | >30 µM | 10.0 µM | 6.1 µM | >30 µM | >30 µM |
| Luminal | MDA MB 134 | 1.9 µM | 2.1 µM | 1.9 µM | 1.9 µM | ND |
| | MCF7 | 3.0 µM | 1.9 µM | 2.7 µM | 3.0 µM | ND |
| | T47D | 6.6 µM | 3.1 µM | 4.2 µM | ND | ND |
| | BT-474 | 4.0 µM | 2.0 µM | 2.0 µM | 6.2 µM | ND |
| | SUM44-PE | 13.5 µM | 4.5 µM | ND | 8.0 µM | ND |
| | MDA MB 361 | >30 µM | 26 µM | 30.0 µM | >30 µM | ND |
| HER2 | SKBR3 | 4.5 µM | 1.3 µM | 2.3 µM | 2.6 µM | ND |
| | HCC1419 | 8.2 µM | 2.9 µM | 3.4 µM | 5.5 µM | ND |

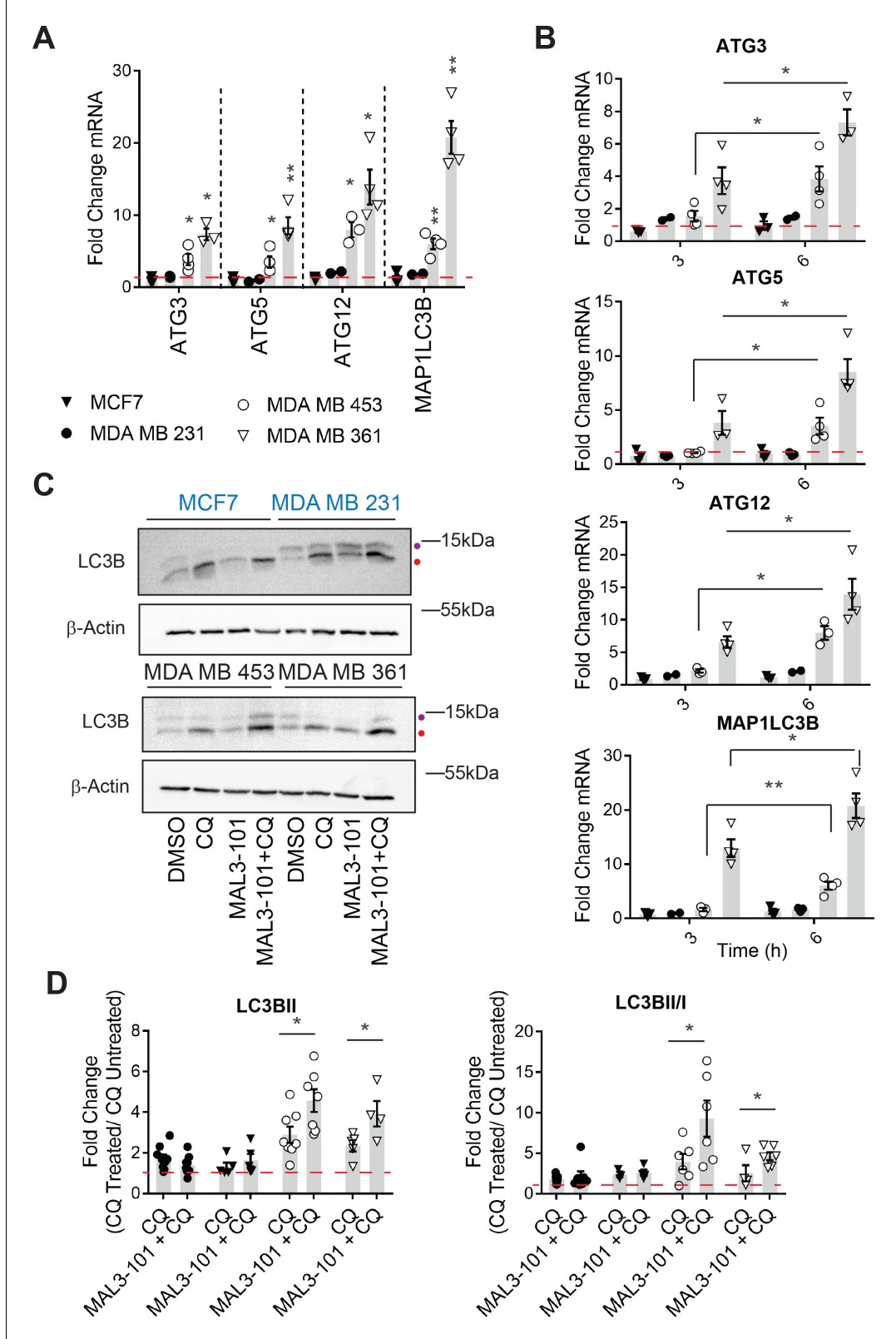

**Figure 4.** Hsp70 inhibition induces higher levels of autophagy in MAL3-101-resistant cells. (**A–B**) The expression of autophagy-associated genes relative to β-actin was detected by qPCR in the presence of DMSO or 12 μM MAL3-101 for 3 and/or 6 hr in sensitive (MCF7 and MDA MB 231, closed symbols) and resistant (MDA MB 453 and MDA MB 361, open symbols) cells. The mean fold change of the indicated mRNAs is shown, ± SEM (n≥3). * denotes p<0.05, ** denotes p<0.005. (**C**) MAL3-101 sensitive (in blue) and resistant (in back) cells were treated for 6 hr with 12 μM MAL3-101 and CQ was added

*Figure 4 continued on next page*

*Figure 4 continued*

during the last 2 hr of the treatment to assess autophagic flux; that is, 'CQ' indicates a 4 hr treatment with DMSO plus a 2 hr acute treatment with CQ. Aliquots of cell lysates were resolved by SDS-PAGE and immunoblotted for LC3B. LC3BI is indicated by a purple dot, and the autophagosome associated form LC3BII is highlighted with a red dot. β-actin serves as a loading control. (D) The fold change in LC3BII levels and the LC3BII/LC3BI ratio between chloroquine (CQ)-treated cells relative to untreated cells was calculated, ± SEM (n≥4). Symbols are defined in panel A. * denotes p<0.05. MAL3-101-sensitive cells are represented with closed symbols, while open symbols indicate MAL3-101-resistant lines.

The online version of this article includes the following source data for figure 4:

**Source data 1.** Raw data for *Figure 4*.

was unchanged regardless of whether MAL3-101 was added alone or in the presence of CQ (*Figure 6A* and *Figure 6—figure supplement 1A*).

To confirm these observations, annexin-V–propidium iodide (PI) staining (*Koopman et al., 1994*; *Vermes et al., 1995*) was used to analyze the percentage of apoptotic cells in the presence or absence of CQ and/or MAL3-101 treatment (*Figure 6B*). Although CQ-mediated inhibition of autophagy had no significant effect on the population of apoptotic cells in both MAL3-101 sensitive and resistant lines, addition of CQ for 2 hr increased the percentage of apoptotic cells when Hsp70 activity was impaired in the resistant cells (MDA MB 453 and MDA MB 361). In contrast, Hsp70 inhibition alone had no effect on the percentage of apoptotic-resistant cells. In addition—and also consistent with the data shown above—MAL3-101 initiated apoptosis in the sensitive (MCF7 and MDA MB 231) cells and the percentage of apoptotic cells were unaltered by CQ (*Figure 6B*). These data establish that combined application of Hsp70 and autophagy inhibitors kills resistant cells.

To better characterize the role of autophagy as a compensatory mechanism upon Hsp70 inhibition, we asked if autophagy induction was sufficient to reduce MAL3-101 sensitivity in the Hsp70 inhibitor-sensitive (MDA MB 231) cell line. The mTOR complex, together with AMPK and HIF1α, are the major autophagy regulators and serve as nutrient, energy, and growth factor sensors (*Hindupur et al., 2015*; *Laplante and Sabatini, 2012*). Thus, we first asked if MAL3-101 sensitive and resistant breast cancer cells exhibited unique sensitivities to the mTOR inhibitor, everolimus (also known as Afinitor), an FDA-approved rapamycin derivative that directly inhibits mTORC1 and indirectly inhibits mTORC2 (*Chen and Zhou, 2020*; *Chiang and Abraham, 2007*; *Meng and Zheng, 2015*). As shown in *Figure 6C*, negligible differences were seen in the sensitivities to everolimus between MAL3-101 sensitive and resistant cell lines (IC$_{50}$23.0 µM and 18.7 µM, respectively). Second, we monitored cell viability in Hsp70 inhibitor sensitive and resistant lines when MAL3-101 was administrated in the presence of sub-maximal doses (2 µM) of everolimus. We found that mTOR inhibition decreased MAL3-101 sensitivity in the MDA MD 231-sensitive line (*Figure 6D*). As expected, no change in MAL3-101 sensitivity was apparent in the presence of everolimus in resistant cells (MDA MB 453, *Figure 6D*). These data are consistent with the model that enhanced proteotoxic stress—via Hsp70 inhibition—and induction of the autophagy pathway protect MAL3-101 sensitive cells when challenged with an Hsp70 inhibitor.

## The UPR and ISR are activated upon Hsp70 inhibition

Defects in chaperone activity cause profound changes in cancer cell proteostasis, leading to induction of the UPR and/or the ISR to modulate folding capacity and attenuate stress (*Cubillos-Ruiz et al., 2017*; *Kim et al., 2015*; *Koromilas, 2015*; *Pakos-Zebrucka et al., 2016*). A consequence of the accumulation of unfolded proteins in the ER is the increased expression of the UPR-related chaperone, BiP (*Bertolotti et al., 2000*; *Walter and Ron, 2011*). In fact, a modest but significant increase in the abundance of the message encoding BiP, HSPA5, was evident in each line examined 6 hr after MAL3-101 treatment, indicating that MAL3-101 mounts an ER stress response in both the MAL3-101 resistant and sensitive lines (*Figure 7A*, right panel), yet leads to cell death only in sensitive cells.

To dissect the contribution of each UPR branch on the effect of MAL3-101, we monitored the transcriptional response in MAL3-101 sensitive and resistant lines by quantifying the abundance of UPR related genes. To investigate the role of Ire1α, the degree to which spliced Xbp1 (sXbp1) accumulated after MAL3-101 administration was measured. A variable but significant accumulation of sXbp1, which is a product of the activated Ire1α endonuclease (*Yoshida et al., 2001a*), was detected after MAL3-101 treatment in both sensitive and resistant cells (*Figure 7A*). MCF7 cells exhibited the

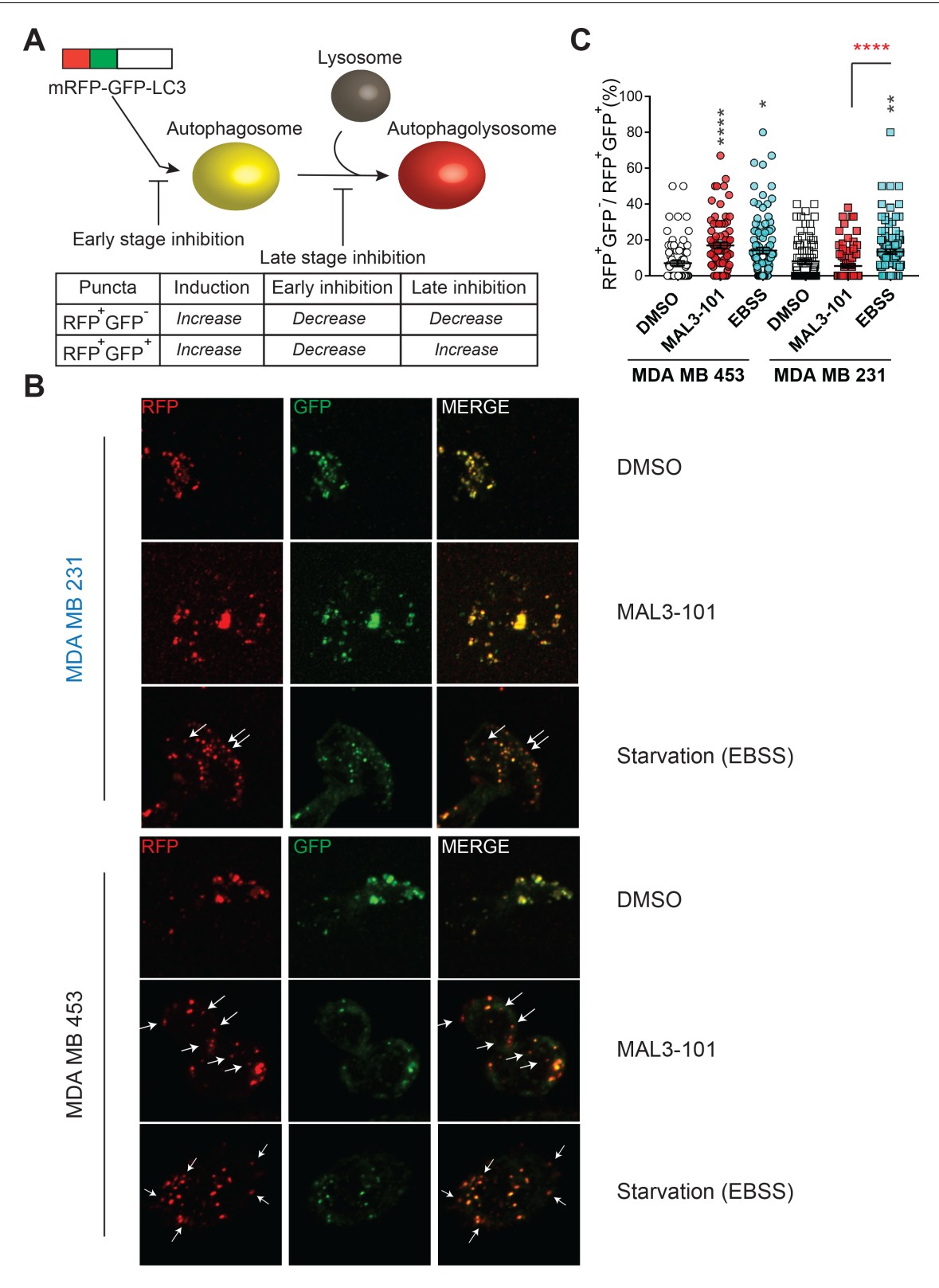

**Figure 5.** Hsp70 inhibition alters autophagic flux in MAL3-101-resistant cells. (**A**) The mRFP-GFP-LC3 reporter monitors autophagic flux. RFP+GFP+ and RFP+GFP- puncta distinguish autophagosomes and autophagolysosomes, thereby allowing for measurements of different stages in the autophagy pathway. (**B**) Confocal images of MAL3-101 sensitive (MDA MB 231, in blue) and resistant (MDA MB 453, in black) clones stably expressing mRFP-GFP-LC3B were treated with 5 μM MAL3-101 or EBSS (starvation media), as a positive control, for 22 hr. White arrows indicate LC3B-positive (acidic)
*Figure 5 continued on next page*

*Figure 5 continued*

compartments (i.e. RFP⁺GFP⁻ puncta). (C) The percentage of RFP⁺GFP⁻ on RFP⁺GFP⁺ puncta per cell is reported for MAL3-101 sensitive (squares) and resistant (circles) lines treated as indicated in B, ± SEM (n≥80). Statistically significant differences between DMSO and cells incubated in starvation media (EBSS) or with MAL3-101 are indicated by black asterisks. Red asterisks represent the statistical significance between the MAL3-101 and EBSS samples. * denotes p<0.05, ** denotes p<0.005, and **** denotes p<0.0001.

The online version of this article includes the following source data for figure 5:

**Source data 1.** Fluorescent puncta quantification.

greatest accumulation of sXbpI upon MAL3-101 treatment, and, interestingly, ERN1 (the gene that encodes Ire1α) copy number is higher in this line (https://depmap.org/portal/). This is further reflected by the high level of increased Ire1α protein expression in these cell lines among the four lines examined (*Figure 7—figure supplement 1A*).

Next, we monitored the ATF6 branch of the UPR by quantifying the abundance of full-length ATF6 after addition of MAL3-101. Upon activation, full-length ATF6 translocates to the Golgi apparatus where it is cleaved to produce an active, soluble ATF6 transcription factor (*Haze et al., 1999*; *Shen et al., 2002*). Thus, a decrease in full-length ATF6 isoform is consistent with enhanced ATF6 cleavage and induction. As shown in *Figure 7B*, the abundance of the ATF6 full-length isoform declined in the presence of DTT, which activates each branch of the UPR, but was unchanged between the MAL3-101 sensitive (MCF7 and MDA MB 231) and resistant (MDA MB 453 and MDA MB 361) breast cancer cells after MAL3-101 treatment.

We then monitored the presence of the DNA damage inducible transcript 3 (DDIT3), which encodes CHOP, as well as the mRNA corresponding to ATF4, which are induced downstream of the p-eIF2α/PERK pathway (*Koromilas, 2015*; *Walter and Ron, 2011*). While ATF4 mRNA was induced to variable levels after MAL3-101 treatment in both the MAL3-101 sensitive and resistant lines, DDIT3 accumulated only in the sensitive (MDA MB 231 and MCF7) cell line (*Figure 7C* and *Figure 7—figure supplement 1B*). This result is consistent with the fact that a pro-apoptotic response is elicited selectively in lines with increased sensitivity to MAL3-101. Because CHOP acts downstream of the p-eIF2α response (*Marciniak et al., 2004*; *Palam et al., 2011*), we then quantified the message encoding another pro-apoptotic factor, PUMA (also known as BBC3), which is transcriptionally activated by ATF4–CHOP (*Galehdar et al., 2010*; *Matus et al., 2013*). Consistent with the data shown in *Figure 7C*, BBC3 (PUMA) mRNA also rose after MAL3-101 treatment in the sensitive but not resistant lines, thereby confirming an apoptotic response in the MAL3-101 sensitive MCF7 and MDA MB 231 lines (*Figure 7—figure supplement 1C*).

To further establish the contribution of the p-eIF2α/PERK pathway in mediating the apoptotic response in select breast cancer cell lines, we measured MAL3-101-dependent changes in select proteins in sensitive and resistant cells. As shown in *Figure 7D* and quantified in *Figure 7—figure supplement 1D*, p-eIF2α accumulated to a similar extent in MAL3-101 sensitive and resistant cells when Hsp70 was inhibited and was indistinguishable when compared to the level observed when cells were treated with DTT. In contrast, and consistent with quantitative PCR data (*Figure 7C* and *Figure 7—figure supplement 1B*), significant CHOP induction was detected only in Hsp70 inhibitor sensitive cells (MCF7 and MDA MB 231) after MAL3-101 addition (*Figure 7E* and *Figure 7—figure supplement 1E*).

Together, the activation of each branch of the UPR—Ire1α, ATF6, and PERK—is consistent with the fact that the sensitive and resistant cells can induce the UPR, but a protective autophagy-dependent response resides selectively in resistant cells. Based on these data, we suggest that resistant cells have been rewired to better counteract proteotoxic insults.

## Ire1α is dispensable for induction of the MAL3-101 stress response

To define the role of Ire1, if any, on MAL3-101-dependent apoptosis, Ire1α was silenced and sensitive (MDA MB 231) and resistant (MDA MB 453) cell survival was examined after MAL3-101 was added. We first noted that less than ~25–35% of Ire1α protein remained 48 hr after siRNA transfection (*Figure 8A and B*), a level similar to that observed in other systems when this message was silenced (*Li et al., 2017*; *Luhr et al., 2019*; *Mu et al., 2008*; *Notte et al., 2015*). In addition—and as anticipated—sXbpI RNA abundance fell when Ire1α was silenced (*Figure 8C*). In contrast, Hsp70 protein abundance was unaffected when Ire1α was silenced (*Figure 8A*), indicating the lack

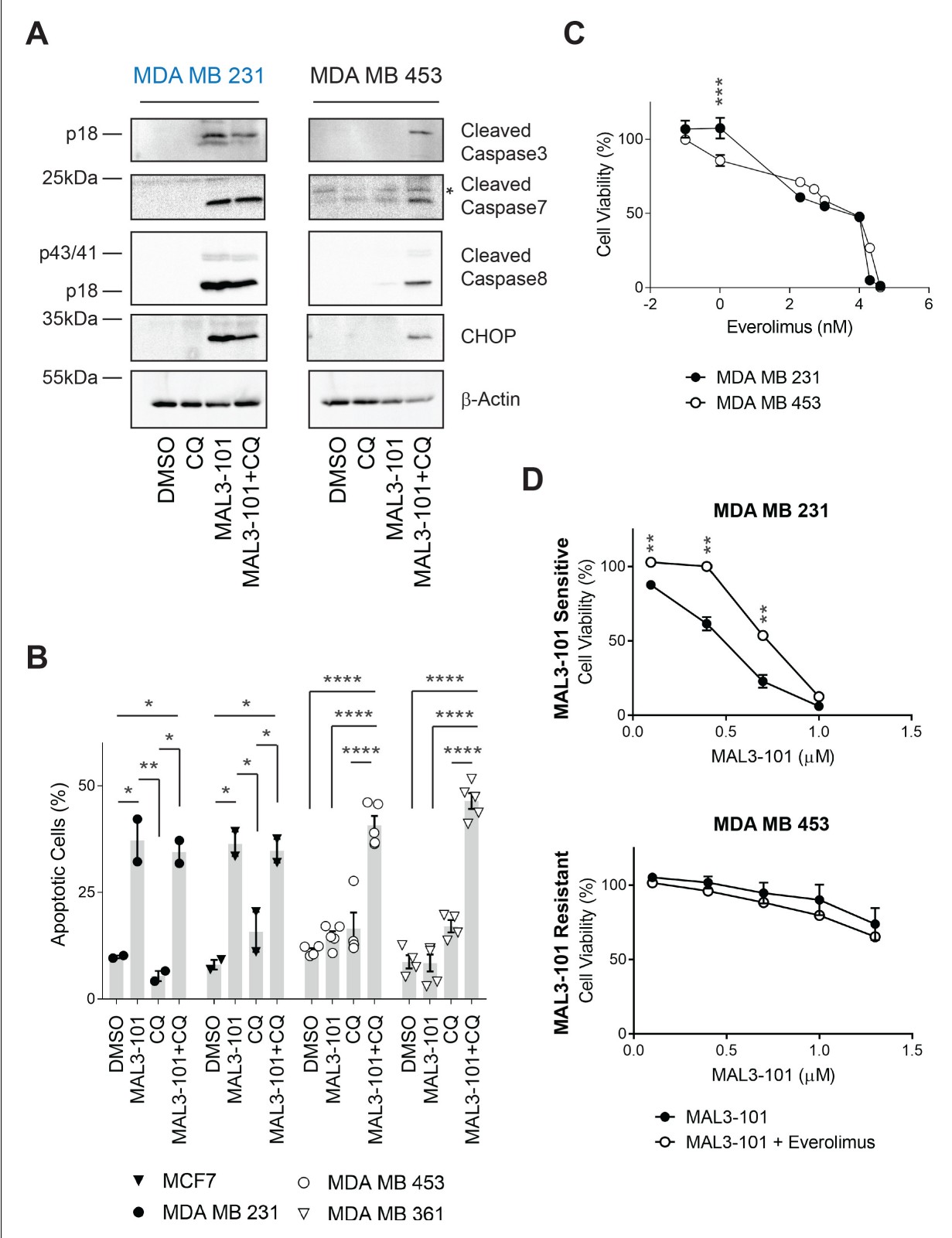

**Figure 6.** Combined inhibition of autophagy and Hsp70 induces apoptosis in MAL3-101 resistant cells. (**A**) MAL3-101 sensitive (in blue) and resistant (in black) cells were treated with 12 μM MAL3-101 for 6 hr and CQ was added during the last 2 hr of the treatment to block autophagy. Lysates prepared from the treated cells were immunoblotted for cleaved caspase-3, caspase-8, and caspase-7, as well as CHOP to monitor apoptosis. β-actin serves as a loading control. * indicates a non-specific band present in the cleaved caspase-7 immunoblot. (**B**) Cells were treated as in panel A before aliquots were

*Figure 6 continued on next page*

*Figure 6 continued*

removed and stained for annexin-V and PI. The sum of the annexin-V positive and the PI and annexin-V double positive cells is represented in the graph and indicated as a percentage of apoptotic cells. MAL3-101-sensitive cells are indicated with closed symbols and MAL3-101-resistant cells are represented by open symbols. The means of independent experiments, ± SEM, are indicated (n=5 for MDA MB 453 and MDA MB 361 cells, and n=2 for MCF7 and MDA MB 231 cells, 50,000 cells were analyzed for each sample). * denotes p<0.05, ** denotes p<0.005, *** denotes p<0.0005, and **** denotes p<0.0001. (C) MAL3-101 sensitive (closed symbols) and resistant (open symbols) cells were treated with increasing doses of everolimus (an mTOR inhibitor) for 72 hr, and cell viability was measured. Data represent the means of three or more independent experiments, ± SEM; * denotes p<0.05. (D) MAL3-101 sensitive and resistant lines (closed and open symbols, respectively) were treated with increasing doses of MAL3-101 in the presence or absence of 2 µM everolimus, as indicated. Cell viability was detected after 72 hr, and data represent the means of two or more independent experiments, ± SEM. * denotes p<0.05, ** denotes p<0.005. Log$_{10}$ concentration of everolimus (C) or MAL3-101 (D) are represented on the x-axis.

The online version of this article includes the following source data and figure supplement(s) for figure 6:

**Source data 1.** Cell viability and apoptotic cell quantification data.

**Figure supplement 1.** MAL3-101 induces apoptosis when autophagy is impaired in resistant cells.

**Figure supplement 1—source data 1.** Raw data for *Figure 6—figure supplement 1*.

of compensatory Hsp70-dependent effects on cellular homeostasis (also see *Figure 1—figure supplement 1*). In addition, the relative levels of p-eIF2α were unaffected by the loss of Ire1α after MAL3-101 was added to either sensitive or resistant cells, as expected since phosphorylation is induced downstream of PERK. Finally, MAL3-101 only increased the levels of CHOP in sensitive cells (MDA MB 231), consistent with data shown in *Figure 7—figure supplement 1E*; nevertheless, the response was again unaffected by the absence of Ire1α (*Figure 8A and D*, compare CTRL and Ire1αi samples).

## Differential effects of eIF2α kinases in breast cancer cell survival

The data outlined above implicate the UPR/ISR in the MAL3-101-dependent response that can lead to cell death. In mammals, four kinases (PERK, GCN2, PKR, and HRI) are linked to the ISR and eIF2α phosphorylation and—as a result to their diverse roles as sensors— myriad stress stimuli can be produced via the ISR (*Ryoo and Vasudevan, 2017*; *Wek, 2018*). PKR is induced upon viral infection and participates in the lysosomal clearance of misfolded proteins (*Darini et al., 2019*; *Pataer et al., 2020*; *Wek, 2018*), GCN2 activation depends on the ribosome P-stalk after amino acid depletion (*Harding et al., 2019*; *Inglis et al., 2019*; *Ye et al., 2010*), and the HRI kinase triggers the ISR when mitochondrial function is altered or heme is depleted. It was also reported that HRI contributes to the general cytosolic unfolded protein response (*Chen and London, 1995*; *Guo et al., 2020*; *Mukherjee et al., 2021*; *Zhang et al., 2019*). Thus, we focused on whether HRI, PERK, PKR, and/or GCN2 were required for MAL3-101 resistance. To this end, we transfected two different siRNAs directed against each kinase. Knockdown efficiency was assessed by immunoblot analysis 72 hr after transfection.

We first investigated the role of HRI during the induction of MAL3-101-mediated apoptosis. siRNA oligo transfection decreased HRI expression levels by 80% in both the MDA MB 231 and MDA MB 453 cells (*Figure 9A and B*), a level comparable to that observed in other systems when HRI was silenced (*White et al., 2018*; *Zhang et al., 2019*). However, HRI knockdown failed to increase Hsp70 abundance, indicating the absence of a compensatory response to MAL3-101 in sensitive and resistant cells. We next measured the apoptotic response upon MAL3-101 treatment in control and HRI knockdown cells. As shown in *Figure 9A* and quantified in *Figure 9C*, HRI silencing modestly increased the accumulation of cleaved caspase-8 and CHOP, but not cleaved caspase-3 (compare CTRL and HRIi samples in *Figure 9A*) in the MAL3-101 sensitive line, MDA MB 231. However, this effect was not statistically significant. Moreover, when HRI expression was blunted in the resistant (MDA MB 453) cells, MAL3-101 failed to induce an apoptotic response. Thus, we conclude that HRI is dispensable for the survival of MAL3-101 resistant cells, but might contribute minimally to MAL3-101-dependent apoptosis in sensitive cells.

Next, we investigated the role of other two ISR sensors, PERK and PKR. Knockdown using targeted siRNAs against PERK and PKR decreased levels by 85% in MDA MB 231 cells and by 85% (PKR) and 70% (PERK) in MDA MB 453 cells (*Figure 10A and B*). Comparable knockdown efficiencies were observed in other systems (*Luhr et al., 2019*; *Mu et al., 2008*; *Pataer et al., 2020*;

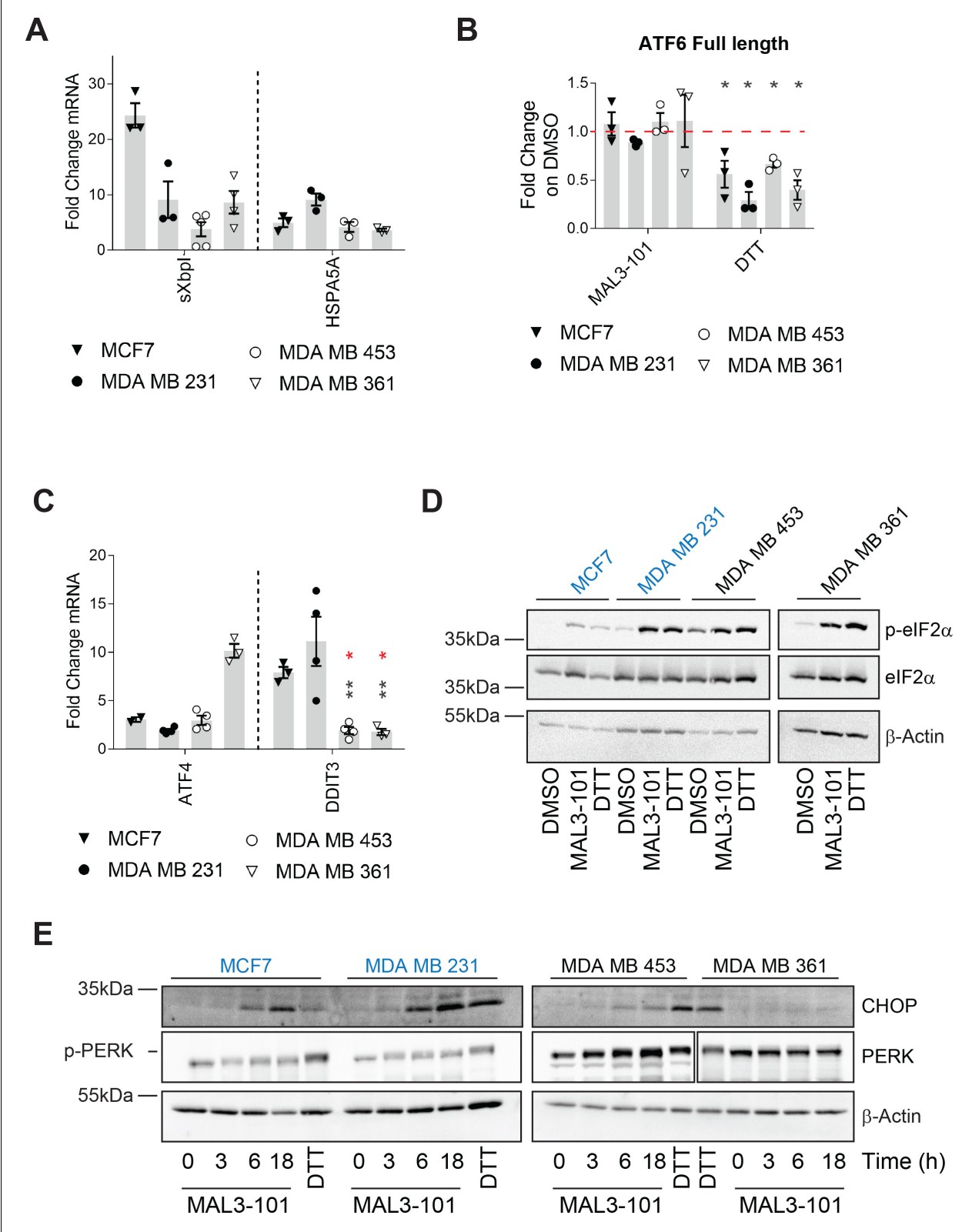

**Figure 7.** Hsp70 inhibition induces the UPR and ISR in breast cancer cells. The relative fold change of (**A**) spliced Xbp1 (sXbpI) and HSPA5A (BiP) mRNA levels relative to β-actin, (**B**) full length ATF6 protein, and (**C**) the ATF4 and DDIT3 (CHOP) transcripts are shown. In part **B**, asterisks represent p<0.05, and in part **C**, black asterisks correspond to p<0.05 between the sensitive (MDA MB 231) (closed circles) and resistant (MDA MB 453 and MDA MB 361) cell lines (open symbols). Red asterisks represent p<0.05 between MCF7 (open triangles) and MDA MB 453 and MDA MB 361 cells (open symbols). In

*Figure 7 continued on next page*

*Figure 7 continued*

all cases, cells were treated with MAL3-101 for 6 hr. In parts **A** and **C**, the bars represent the amounts of the indicated transcripts in treated cells relative to the DMSO control. In part **B**, a 2 hr DTT treatment was used as a positive control. Note that the levels of induction by DTT and MAL3-101 are similar. Data depict ± SEM (n≥3). (**D**) Immunoblots for eIF2α and its phosphorylated isoform in MAL3-101 sensitive (in blue) and resistant (in black) cell lines treated with 12 µM MAL3-101 or DMSO for 6 hr are shown. (**E**) MAL3-101 sensitive (in blue) and resistant (in black) cells were treated with 12 µM MAL3-101 or DMSO for the indicated time and lysates were immunoblotted for CHOP and PERK expression. In part D and E, a 2 hr DTT treatment was used as a positive control and β-actin serves as a loading control.

The online version of this article includes the following source data and figure supplement(s) for figure 7:

**Source data 1.** Source data for *Figure 7*.
**Figure supplement 1.** eIF2α is phosphorylated in response to MAL3-101.
**Figure supplement 1—source data 1.** UPR marker quanficiation raw data.

*Tang et al., 2019*). Nevertheless, to show that these knockdown conditions were sufficient, we found that p-eIF2α and CHOP levels fell when PERK was silenced compared to control transfected samples in sensitive and resistant cells treated with the UPR stressor tunicamycin (*Figure 10—figure supplement 1B*).

Having validated the knockdown conditions, we next measured the induction of apoptotic markers under control or knockdown conditions and in the presence or absence of MAL3-101. As shown in *Figure 10A* and quantified in *Figure 10—figure supplement 1A* (see relative 'CTRL' levels), PKR silencing had no effect on the accumulation of cleaved caspase-3, cleaved caspase-8, or CHOP, but in contrast reduced PERK activity decreased the levels of the apoptotic markers in sensitive (MDA MB 231) cells in the presence of MAL3-101. However, when PERK or PKR was silenced in the resistant (MDA MB 453) cells, an apoptotic response was absent when MAL3-101 was added. To confirm PERK activation upon MAL3-101 treatment, its phosphorylation status (p-PERK) was monitored by examining whether mobility was slowed after SDS-PAGE and immunoblotting (*Figure 10C*). We found that PERK mobility was reduced to a similar extent in MAL3-101 sensitive and resistant cells when Hsp70 was inhibited with MAL3-101 or when cells were treated with tunicamycin. These data confirm that PERK is activated in both MAL3-101 sensitive and resistant cells upon Hsp70 inhibition, yet PERK is only responsible for drug-dependent apoptosis in the sensitive cell lines.

To confirm the contribution of PERK in MAL3-101-mediated apoptosis in sensitive cells, we pretreated sensitive and resistant cells with a PERK inhibitor, GSK-2606414 ('GSK'), prior to MAL3-101 administration (*Axten et al., 2012*). We then monitored the induction of apoptosis in the presence or absence of PERK inhibition when Hsp70 activity was impaired (MAL3-101 addition) or in control (DMSO) conditions (*Figure 10D* and *Figure 10—figure supplement 1D*). In line with our data, above, on the role played by PERK in initiating apoptosis in the sensitive cell line, GSK reduced MAL3-101-mediated apoptosis in MDA MB 231 cells upon Hsp70 inhibition (*Figure 10D* and *Figure 10—figure supplement 1D*). In contrast, and also in line with the data presented above, PERK activity was dispensable for triggering MAL3-101-driven cell death in the resistant MDA MB 453 cell line.

Finally, we examined the role of GCN2 in breast cancer cell survival upon Hsp70 inhibition. Only ~25% of GCN2 remained after knockdown in either cell line (*Figure 11A and B*), a level comparable to that observed in other systems in which the contribution of this kinase on cellular stress responses was analyzed (*Rajanala et al., 2019*; *Shi et al., 2019*). We subsequently measured the accumulation of CHOP, cleaved caspase-3, and cleaved caspase-8 in MAL3-101-sensitive cells (MDA MB 231) after GCN2 was silenced, but the levels of these reporters were unaltered (*Figure 11A and C*). Notably, GCN2 knockdown failed to increase Hsp70 levels, indicating that a compensatory response to MAL3-101 was absent in the sensitive cells.

We next asked if GCN2 plays a role in the pro-survival response in MDA MB 453 resistant cells. In contrast to the modest effect of PERK and PKR on eIF2α phosphorylation, p-eIF2α levels significantly decreased when GCN2 was silenced in these cells in the presence of MAL3-101 (*Figure 11A and C*, see relative 'CTRL' levels). GCN2 knockdown also induced apoptosis when Hsp70 activity was impaired with MAL3-101 (*Figure 11A and C*). Interestingly, the phosphorylation of Thr-899 in the activation-loop of GCN2 was unaltered by MAL3-101 addition (pGCN2, *Figure 11—figure supplement 1A*). This result suggests that other sites might be phosphorylated when Hsp70 activity is

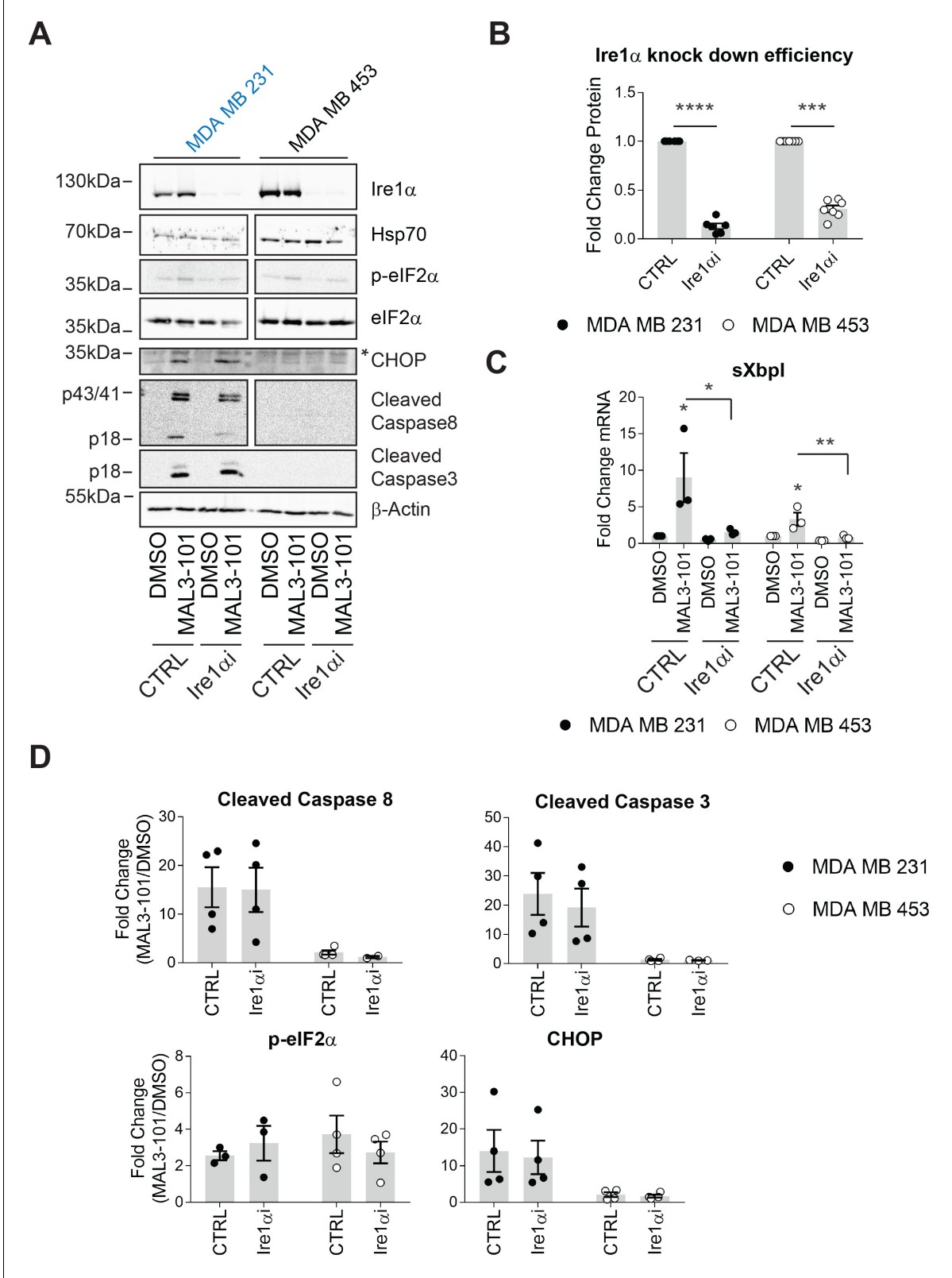

**Figure 8.** The MAL3-101 response is Ire1-independent. (**A**) MAL3-101-sensitive cells (in blue) and MAL3-101-resistant cells (in black) were transfected with a control or an Ire1α-targeted siRNA. Forty-eight hr after transfection, the cells were treated with 12 μM MAL3-101 or DMSO for 6 hr, and lysates were immunoblotted for the indicated proteins. * indicates a nonspecific band in the CHOP immunoblot. β-actin served as a loading control. (**B**) Ire1α knockdown efficiency was measured by immunoblot in MAL3-101 sensitive (closed symbols) and resistant (open symbols) cells and the fold decrease is

*Figure 8 continued on next page*

*Figure 8 continued*

shown compared to the siRNA control, ± SEM (n=7). *** denotes p<0.0005 and **** denotes p<0.00001. (C) The means, ± SEM, of the relative level of sXbpI message were calculated after qPCR in control MAL3-101 sensitive (closed symbols) and resistant (open symbols) cells transfected with a control siRNA or an siRNA directed against Ire1α, which were then treated with 12 μM MAL3-101 or DMSO for 6 hr. * denotes p<0.05 and ** denotes p<0.005. (D) The means of the corresponding fold increase in MAL3-101 sensitive (closed symbols) and resistant (open symbols) cells of the indicated apoptotic markers and p-eIF2α were obtained in control and Ire1α siRNA transfected cells which were then treated with MAL3-101 or DMSO for 6 hr, ± SEM (n=4).

The online version of this article includes the following source data for figure 8:

**Source data 1.** Source data for *Figure 8*.

impaired. By analogy, a previous report hinted that other residues might be phosphorylated in response to ribosome P-stalk-mediated activation of GCN2, and multiple GCN2 phosphorylation events were detected upon amino acid starvation which were independent of Thr-899 phosphorylation (*Harding et al., 2019*). In addition, the increased accumulation of CHOP upon MAL3-101 treatment when GCN2 expression was silenced suggests that apoptosis might depend on the activation of another ISR sensor (*Figure 11A and C*) in resistant (MDA MB 453) cells. Regardless, these results position the GCN2-dependent phosphorylation of eIF2α as a trigger for protection against compromised proteostasis and the induction of a pro-survival degradation pathway, that is, autophagy, in MAL3-101-sensitive cells.

To confirm the role of the ISR in MAL3-101-associated apoptosis, we monitored apoptotic induction in the presence or absence of the ISR inhibitor, ISRIB (*Halliday et al., 2015*; *Rabouw et al., 2019*; *Sidrauski et al., 2015*; *Zyryanova et al., 2018*). ISRIB is an allosteric inhibitor of the downstream eIF2α target, eIF2B, and ultimately blocks the ISR (*Zyryanova et al., 2018*). We found that ISRIB had no effect on p-eIF2α accumulation (*Figure 11—figure supplements 1A and B*, compare 'MAL3-101' and 'MAL3-101+ISRIB'), yet ISRIB treatment decreased CHOP, cleaved caspase-3, and cleaved caspase-8 accumulation in sensitive (MDA MB 231) cells after MAL3-101 was added (*Figure 11—figure supplements 1A and C*, see relative 'MAL3-101' levels). When the ISR was inhibited with ISRIB in the resistant MDA MB 453 cells, only a modest accumulation of cleaved caspase-8 was induced when MAL3-101 was present (*Figure 11—figure supplements 1A and C*). Thus, taken together with the data presented above, these results identify the ISR as an apoptotic trigger in sensitive breast cancer cell lines when proteostasis is compromised.

## GCN2 regulates the autophagy pathway in MAL3-101-resistant breast cancer cells

Hsp70 inhibition induces the accumulation of p-eIF2α to a similar extent in MAL3-101 sensitive and resistant breast cancer lines, but the eIF2α kinases (PERK versus GCN2) and the final cellular outcomes (apoptosis versus autophagy, respectively) are distinct: PERK knockdown blunted the death response in sensitive cells after MAL3-101 treatment, whereas the silencing of GCN2 favored the death of resistant cells after the same treatment. To better define the roles of the PERK and GCN2 kinases in the induction of the compensatory (i.e., autophagy) response in resistant cells, we monitored autophagy activation in control and PERK and GCN2 silenced cells in the presence or absence of MAL3-101. In this experiment, CQ was added during the last 2 hr of MAL3-101 treatment to block autophagolysosome-dependent degradation, which allowed us to focus on autophagic flux under each condition.

First, we confirmed that autophagy—as measured by both the level of LC3BII and the LC3BII:LC3BI ratio—was present at a higher level in resistant (MDA MB 453) compared to sensitive cells (MDA MB 231) transfected with a control siRNA and were either treated or left untreated with MAL3-101 (*Figure 12A*, and data quantified in *Figure 12—figure supplement 1A,B*; see relative 'CTRL' levels). Next, we found that the accumulation of the autophagy marker LC3BII and the LC3BII:LC3BI ratio decreased in CQ-treated resistant cells (MDA MB 453) in which PERK expression was blunted compared to control cells. This result suggests that PERK enhances basal autophagic flux in MAL3-101-resistant cells. However, MAL3-101 treatment still increased the amount of LC3BII as well as the LC3BII:LC3BI ratio, suggesting that PERK is dispensable for MAL3-101-mediated autophagy induction in the resistant cells. In contrast, and as anticipated, there was no significant change in LC3BII accumulation and the LC3BII:LC3BI ratio between control and PERK siRNA

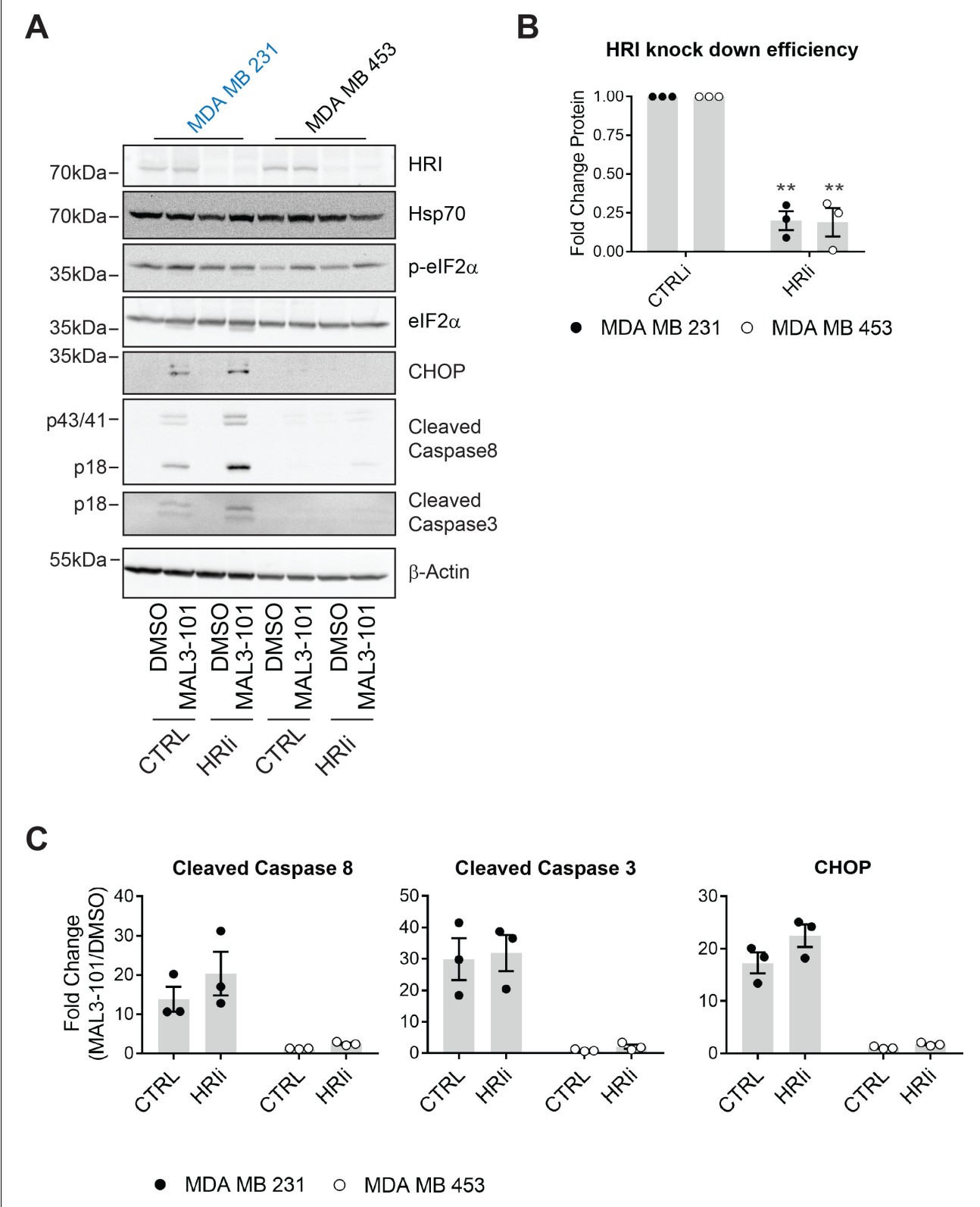

**Figure 9.** HRI is largely dispensable for the survival of breast cancer cells challenged with MAL3-101. (**A**) MAL3-101 sensitive (in blue) and resistant (in black) lines were transfected with a control siRNA or an siRNA oligo directed against HRI, and 72 hr post transfection cells were treated with DMSO or 12 µM MAL3-101 for 6 hr. Lysates were prepared for immunoblot analysis. β-actin serves as a loading control. (**B**) HRI knockdown efficiency was measured by immunoblot 72 hr post siRNA transfection in MAL3-101 sensitive (MDA MB 231, closed circles) and resistant (MDA MB 453, open circles)

*Figure 9 continued on next page*

*Figure 9 continued*

cell lines. The relative fold change of HRI protein levels in knockdown to control samples is plotted ± SEM. ** denotes p<0.005. (**C**) The corresponding fold increase of cleaved caspase-8, cleaved caspase-3, and CHOP in MAL3-101 sensitive (closed symbols) and resistant (open symbols) cells was measured after treatment with control, or HRI siRNAs. The fold change is shown relative to the DMSO control, ± SEM (n=3).

The online version of this article includes the following source data for figure 9:

**Source data 1.** Source data for *Figure 9*.

transfected sensitive (MDA MB 231) cells in the presence of MAL3-101 (*Figure 12A*, *Figure 12—figure supplement 1A,B*; compare 'MAL3-101' CTRL with 'MAL3-101' PERKi).

Second, we asked if GCN2 activates autophagy in the presence or absence of MAL3-101. In resistant cells (MDA MB 453), the accumulation of the autophagy marker LC3BII and the LC3BII:LC3BI ratio decreased when GCN2 was silenced and when Hsp70 was inhibited, suggesting that GCN2 is critical for MAL3-101-mediated autophagy induction (*Figure 12B*, *Figure 12—figure supplement 1B*; note decreased signal for 'MAL3-101', 'GCN2i'). On the contrary, no change in autophagy marker abundance was detected in resistant cells (MDA MB 453) treated only with CQ, suggesting that the basal autophagy flux is unaltered by GNC2 silencing (*Figure 11B*, *Figure 12—figure supplement 1B*). Autophagy was also unaltered in sensitive cells (MDA MB 231) upon GCN2 protein ablation (*Figure 12B*, *Figure 12—figure supplement 1A*; 'GCN2i'). These data confirm the role of GCN2 in autophagy induction when proteotoxic stress rises in resistant cells.

PERK partially compensates for autophagy induction in GCN2$^{-/-}$ MEF cells under stress conditions (*Ye et al., 2010*). Therefore, to better define the role of PERK and GCN2 in MAL3-101-mediated autophagy in sensitive and resistant cells, we performed a PERK-GCN2 double knockdown and compared both LC3BII accumulation and the LC3BI:LC3BII ratio to the control (*Figure 12C*). Genetic ablation of the two eIF2α kinases impaired basal and MAL3-101-mediated autophagy in MAL3-101 resistant (MDA MB 453), but not in sensitive (MDA MB 231) cells (*Figure 12C*, *Figure 12—figure supplement 1A and B*; 'PERKi GCN2i'). Moreover, the knockdown of both PERK and GCN2 showed an additive effect as LC3BII accumulation along with LC3BI to LC3BII conversion in resistant cells (MDA MB 453) decreased even more compared to the single knockdowns, at least in the absence of MAL3-101 (*Figure 12C* and *Figure 12—figure supplement 1B*). Taken together, our data demonstrate that PERK oversees autophagy in the absence of Hsp70 inhibition, but GNC2 is fundamental for MAL3-101-dependent autophagy in resistant cells.

## Induction of the ER stress response and protein folding pathway genes correlate with Hsp70 inhibition in sensitive cells

To better define the difference in the sensitivities of breast cancer cells to MAL3-101, we analyzed publicly available RNA sequencing (RNAseq) databases from the Broad Institute Cancer Cell Line Encyclopedia (CCLE) (*Barretina et al., 2012*) and RNAseq data published in prior work (*Marcotte et al., 2016*). We also evaluated the breast cancer cell proteome using the public available RPPA datasets (*Marcotte et al., 2016*; *Figure 13A and B*). Interestingly, Hsp70 inhibitor sensitivity failed to correlate with the transcriptional profiles of the 14 breast cancer cells examined in our study (*Figure 13A*). On the contrary, the RPPA dataset analysis revealed that resistant breast cell lines (MDA MB 453, MDA MB 361 and HCC38) have higher levels of activated AMPK (AMPKα-pT172) and of the ribosomal S6 kinase (p70-S6K-pT389) (*Figure 13B*). These proteins, respectively, are direct autophagy inducers in response to energy availability and regulate translation elongation downstream of mTOR signaling (*Hindupur et al., 2015*). Increased pT172 AMPK levels also suggest that MAL3-101-resistant cells exploit AMPK activation to modulate their metabolic state to maintain an adequate energy balance even under proteotoxic stress conditions (*Garcia and Shaw, 2017*). In agreement with this hypothesis, both increased pT172 AMPK and p70-S6K-pT389 levels accumulate upon glucose starvation in HEK293 cells when autophagy is induced (*Kim et al., 2011*). Thus, MAL3-101-resistant cells are primed to modulate metabolism (i.e. inducing autophagy) to counteract imbalanced proteostasis.

To further define the pathways involved in the differential MAL3-101 response between sensitive and resistant cells, we also performed an RNAseq analysis after sensitive (MDA MB 231) and resistant (MDA MB 453) cells were treated with MAL3-101 or the DMSO control. MAL3-101 treatment

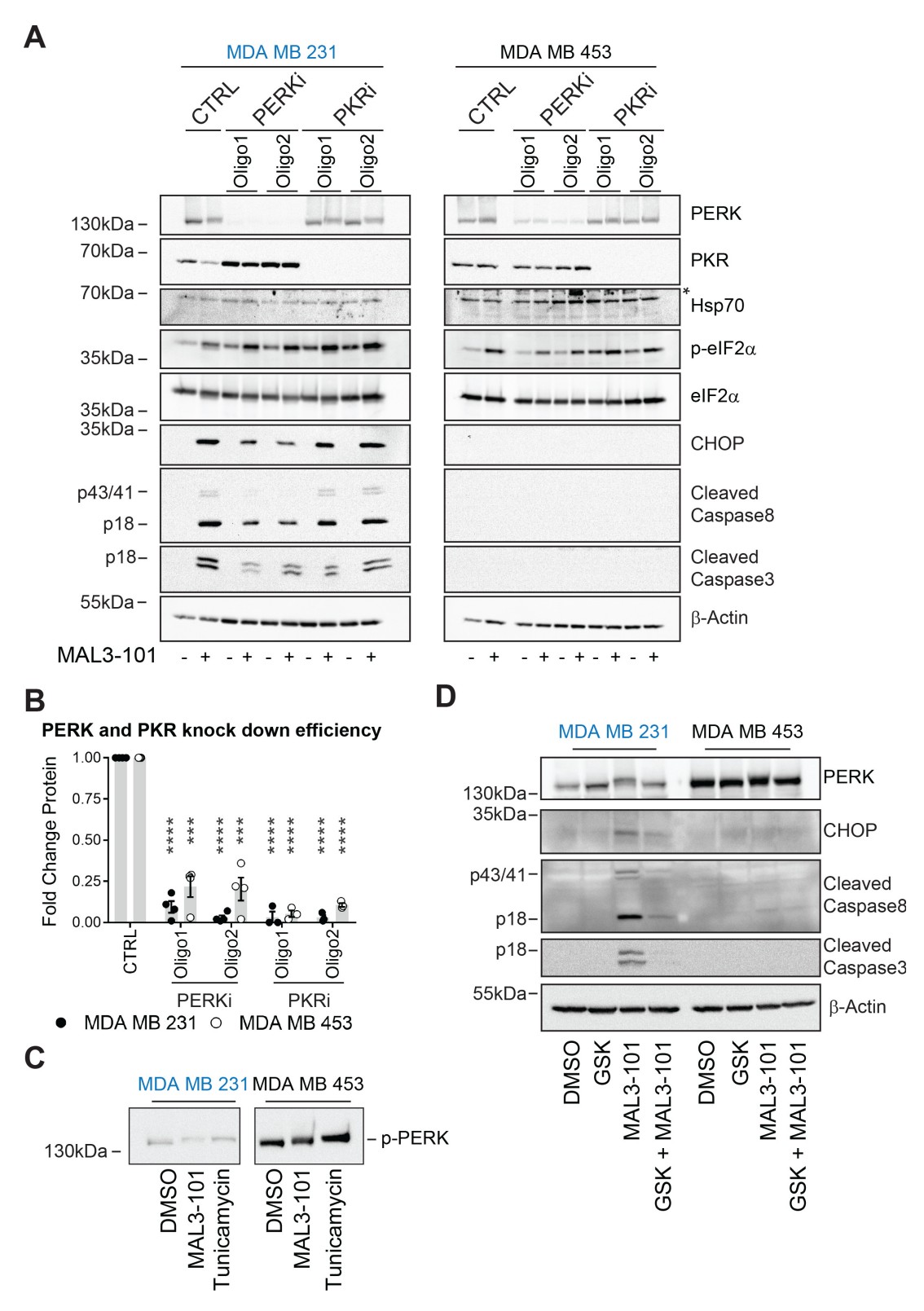

**Figure 10.** PERK is required for apoptosis induction in Hsp70 inhibitor sensitive cells. (**A**) MAL3-101 sensitive (in blue) and resistant (in black) cells lines were transfected with a control siRNA or two different siRNAs that were directed against PERK or PKR. 72 hr post transfection cells were treated with DMSO or 12 μM MAL3-101 for 6 hr and lysates were prepared for immunoblot analysis. β-actin serves as a loading control, and the asterisk denotes a non-specific band in the Hsp70 blot. (**B**) PERK and PKR knockdown efficiency were measured by immunoblot 72 hr post siRNA transfection in MAL3-101

*Figure 10 continued on next page*

*Figure 10 continued*

sensitive (MDA MB 231, closed circles) and resistant (MDA MB 453, open circles) cell lines. The relative fold change of PERK and PKR protein levels in knockdown to control samples is plotted ± SEM. *** denotes p<0.0005 and **** denotes p<0.00001. (C) MAL3-101 sensitive and resistant cells were treated with 12 µM MAL3-101, DMSO for 6 hr or with 10 µg/µl tunicamycin for 3 hr. PERK mobility was analyzed by immunoblot. (D) MAL3-101 sensitive (in blue) and resistant (in black) cells lines pre-treated with GSK-2606414 (GSK, 2 µM) for 2 hr prior to addition of DMSO or 12 µM MAL3-101 for 6 hr. Lysates were prepared for immunoblot analysis. β-actin serves as a loading control.

The online version of this article includes the following source data and figure supplement(s) for figure 10:

**Source data 1.** PERK and PKR raw quantification.
**Figure supplement 1.** PERK knockdown reduces apoptosis.
**Figure supplement 1—source data 1.** Source data for *Figure 10—figure supplement 1*.

had a minor impact on the global transcriptional profile of the resistant cells (MDA MB 453, samples A), whereas significantly more variability in the transcriptional profile of MDA MB 231 cells was detected when comparing the MAL3-101 and DMSO-treated samples (samples B, *Figure 13C*). In agreement with the global transcriptomic analysis, above, hierarchical clustering of the samples based on differential gene analysis revealed that MAL3-101-treated sensitive cells (MDA MB 231) cluster with resistant MDA MB 453 cells (*Figure 13D*), while no change was visible in between MAL3-101 and DMSO treated resistant cells (MDA MB 453). These transcriptomic analyses suggest that MAL3-101 sensitivity is more tightly linked to changes in the cancer cell proteome and metabolome compared to the transcriptional profiles. This hypothesis will be pursued in future work. In fact, the top 35 genes differentially regulated between sensitive and resistant cells failed to correlate with any specific molecular mechanism or biological process (*Figure 13D*).

Finally, to identify other factors that might dictate ISR-related circuits in MAL3-101 sensitive cells, we compared the two sample cohorts in MDA MB 231 cells to assess the presence of genes that may drive MAL3-101 sensitivity. We found that protein folding and the ER stress response pathways were upregulated in MAL3-101-sensitive cells, MDA MB 231, when Hsp70 activity was inhibited (*Figure 13E*). These data confirm the key role of the UPR and, in particular, of the PERK branch of this pathway in dictating the fate of MAL3-101 sensitive cells when Hsp70 activity is inhibited.

## Discussion

Protein homeostasis, that is, proteostasis, balances metabolic demands via protein synthesis with protein quality control, which clears the cell of potentially toxic misfolded proteins (*Balch et al., 2008*; *Jayaraj et al., 2020*; *Luh and Bertolotti, 2020*). In some cancer cells, proteostasis modulation is essential for cell survival, rapid rates of cell division, and metastasis (*Nam and Jeon, 2019*; *Wang and Kaufman, 2014*). In this work, we utilized an established allosteric inhibitor of Hsp70-mediated proteostasis, MAL3-101 (*Gestwicki and Shao, 2019*; *Huryn et al., 2011*), to explore how cells resist a proteostasis collapse. This goal was justified based on Hsp70's prominent role in almost every branch of the proteostasis pathway. As a model system for these studies, we used breast cancer cells since Hsp70 is overexpressed in metastatic breast cancer, prevents apoptosis, and contributes to the limited efficacy of Hsp90-based treatments in clinical trials (*Garrido et al., 2006*; *Goloudina et al., 2012*; *Guo et al., 2005*; *Powers et al., 2010*; *Powers et al., 2013*). Ultimately, our analysis identified a range of sensitivities to MAL3-101 among the tested cell lines. The lines included the main breast cancer subtypes, confirming the heterogeneity of this type of cancer. The focus on Hsp70 in these cancer-derived cells lines also highlights the importance of precision medicine strategies based on tumor proteostasis (*Naito and Urasaki, 2018*).

Prior work indicated that MAL3-101 primarily targets cytosolic Hsp70, but the activity of the ER resident Hsp70 chaperone, BiP, is also somewhat affected by the drug (*Fewell et al., 2004*; *Patham et al., 2009*; *Sabnis et al., 2016*; *Sannino et al., 2018*; *Singh et al., 2020*). Therefore, effects on both cytosolic stress responses as well as the UPR/ISR were anticipated, as shown in the current study. Specifically, we find that Hsp70 inhibition initiates a p-eIF2α-mediated stress response in Hsp70 inhibitor sensitive and resistant lines, suggesting that cancer cells attempt to ameliorate stress by modulating their transcriptomes and proteomes to survive and proliferate. Yet, apoptosis can result if key changes in a new proteostasis set-point cannot be attained. By using archived, well established cancer cell lines, we capitalized on naturally occurring variations in these responses.

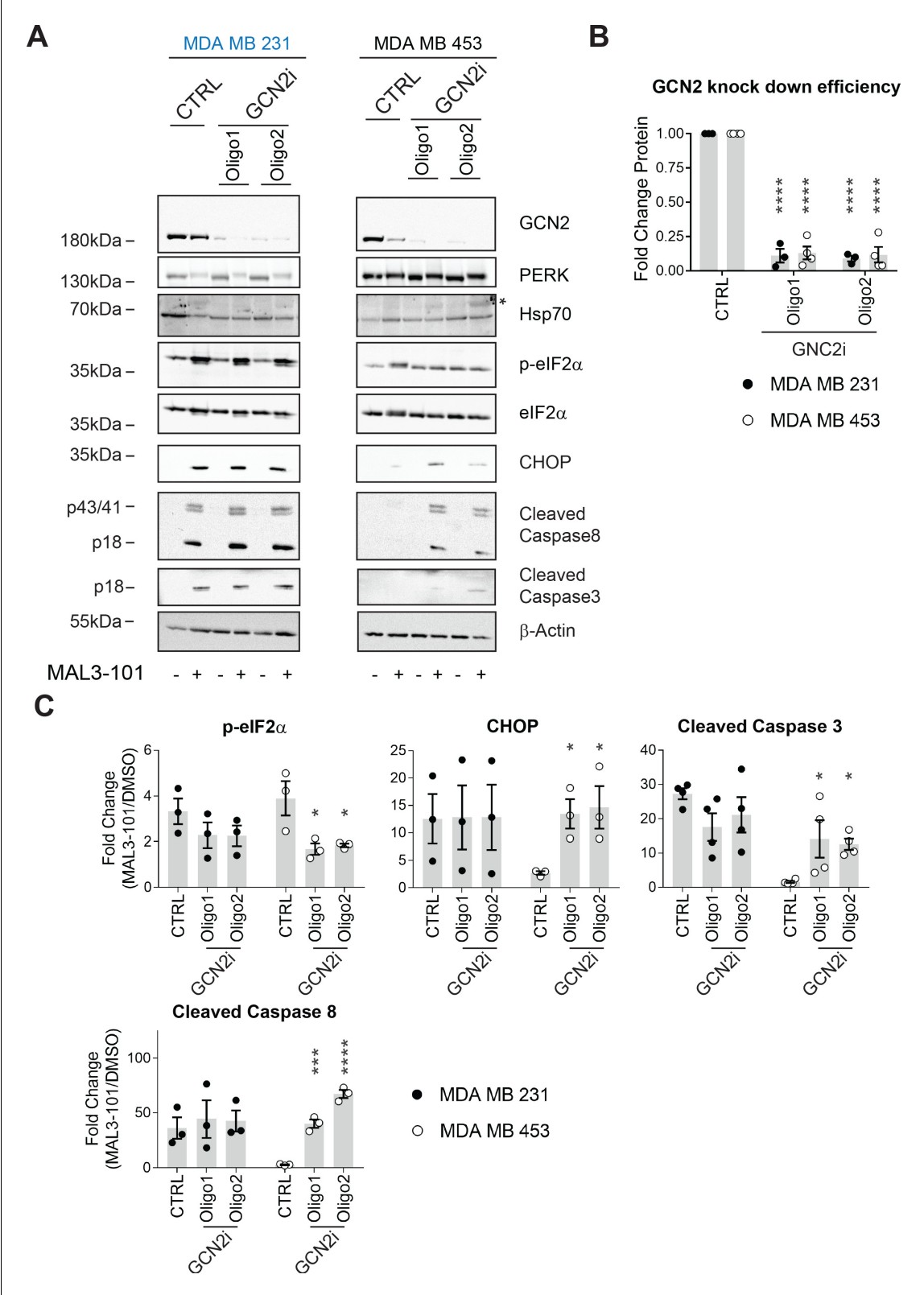

**Figure 11.** GCN2 is required for resistant cell survival when challenged with MAL3-101. (**A**) MAL3-101 sensitive (MDA MB 231, in blue) and resistant (MDA MB 453, in black) cells were transfected with a control siRNA or two different siRNAs directed against GCN2, and 72 hr post transfection cells were treated with DMSO or 12 μM MAL3-101 for 6 hr. Lysates were immunoblotted to detect the indicated proteins. A non-specific band in the Hsp70 immunoblot is indicated with an asterisk (*), and β-actin served as a loading control. (**B**) GCN2 knockdown efficiency was measured by immunoblot 72

*Figure 11 continued on next page*

Figure 11 continued

hr post siRNA transfection in MAL3-101-sensitive (MDA MB 231, closed circles) and MAL3-101-resistant (MDA MB 453, open circles) lines. The relative GCN2 fold change in knockdown to control samples is plotted, ± SEM (n≥3); **** denotes p<0.00001. (**C**) The corresponding fold increase of the indicated apoptotic markers and p-eIF2α in MAL3-101-sensitive (MDA MB 231, closed symbols) and MAL3-101-resistant (MDA MB 453, open symbols) cells treated with a control or siRNAs directed against GCN2 and treated with DMSO or MAL3-101 is shown, ± SEM (n≥3) * denotes p<0.05, *** denotes p<0.0005 and **** denotes p<0.00001.

The online version of this article includes the following source data and figure supplement(s) for figure 11:

**Source data 1.** Source data for *Figure 11*.
**Figure supplement 1.** ISR orchestrates a cell survival response when Hsp70 is inhibited.
**Figure supplement 1—source data 1.** Source data for *Figure 11—figure supplement 1*.

Indeed, prior work indicated that the ISR/UPR and eIF2α phosphorylation can regulate cancer cell fate, depending on the nature of the stress and metabolic state (*Blagden and Willis, 2011*; *Chiti and Dobson, 2006*; *Darini et al., 2019*; *Guo et al., 2017*; *Kim et al., 2015*; *Koromilas, 2015*; *Kouroku et al., 2007*; *Nam and Jeon, 2019*; *Pakos-Zebrucka et al., 2016*; *Travers et al., 2000*; *Yoshida et al., 2001a*; *Yoshida et al., 2001b*). In this study, we discovered that p-eIF2α signaling elicits an apoptotic response at both the mRNA and protein levels in MAL3-101 sensitive cells. On the other hand, MAL3-101-resistant cells exploit p-eIF2α signaling to augment a specific protein degradation—autophagy—which overcomes the rise in toxic proteins and prevents cell death. In accordance with these findings, induction of autophagy via mTOR inhibition reduced the survival of MAL3-101-sensitive cells in the presence of MAL3-101. The role of autophagy as a pro-survival mechanism is supported further by other studies in which this pathway was shown to confer resistance to chemotherapy, immunotherapy, and HER2- and ER-targeted therapies in breast cancer (*Martin et al., 2009*; *Robainas et al., 2017*; *Samaddar et al., 2008*; *Schoenlein et al., 2009*; *Sui et al., 2013*; *Tracey et al., 2020*; *Vazquez-Martin et al., 2009*; *Zambrano and Yeh, 2016*). In addition, we previously reported that lab-generated MAL3-101-resistant rhabdomyosarcoma cells, which were obtained by long-term dose escalation, were partially resensitized to MAL3-101 if autophagy was simultaneously inhibited (*Sannino et al., 2018*). Together, our study supports the importance of monitoring autophagy efficiency in various cancers in parallel to the development of specific autophagy inhibitors that can be administrated alone or in combination with approved secondary therapies (*Amaravadi et al., 2011*; *Chude and Amaravadi, 2017*; *Piffoux et al., 2021*; *Rebecca et al., 2019*; *Shi et al., 2017*).

While the current work employed a cancer cell model, the survival of all cells during homeostatic changes requires proteostasis adaptation through modulation of the ISR and UPR (*Grandjean and Wiseman, 2020*; *Hetz et al., 2020*). What is a matter of investigation—and can be specifically investigated by genetic approaches—is the role of each stress sensor in cell survival. To our knowledge, our work highlights the first example in which different ISR sensors dictate unique fates in a specific cell type. While PERK primarily initiates a pro-apoptotic response in MAL3-101 sensitive cells, GCN2 oversees an autophagy-mediated pro-survival mechanism in resistant cells when Hsp70 is inhibited. Specifically, GCN2 knockdown blunted the MAL3-101-dependent induction of autophagy, leading to CHOP accumulation and cell death (*Figures 10* and *11*). Based on our results, we propose a model (*Figure 14*) in which Hsp70 inhibition increases the accumulation of unfolded proteins that trigger the ISR. Because GCN2 regulates the amino acid response in a manner that depends on the tRNA-sensing ribosomal P-stalk (*Harding et al., 2019*; *Harding et al., 2003*; *Inglis et al., 2019*; *Nakamura et al., 2018*; *Ye et al., 2010*), we reason that Hsp70 inhibition favors the accumulation of unfolded ubiquitinated proteins (*Figure 2A*; *Sannino et al., 2018*), thereby limiting amino acid availability and protein translation. Numerous other examples exist in which a rise in unfolded proteins, by inhibiting the ubiquitin-proteasome pathway, similarly activates a starvation response (*Fan et al., 2018*; *Kocaturk and Gozuacik, 2018*; *Wojcik, 2013*).

In the resistant cells, we also found that induction of GCN2/p-eIF2α signaling increases the levels of autophagy-related genes when Hsp70 is inhibited, favoring breast cancer cell survival (*Figure 12*). Accordingly, the GCN2/ATF4 pathway was reported to play a role in the transcriptional activation of autophagy genes such as MAP1LC3B, ATG3, ATG12, and Becn1 upon amino acid starvation (*B'chir et al., 2013*). In line with the contribution of GCN2 signaling in balancing amino acid availability in resistant cells, asparagine synthetase expression (ASNS) is induced in MAL3-101-resistant

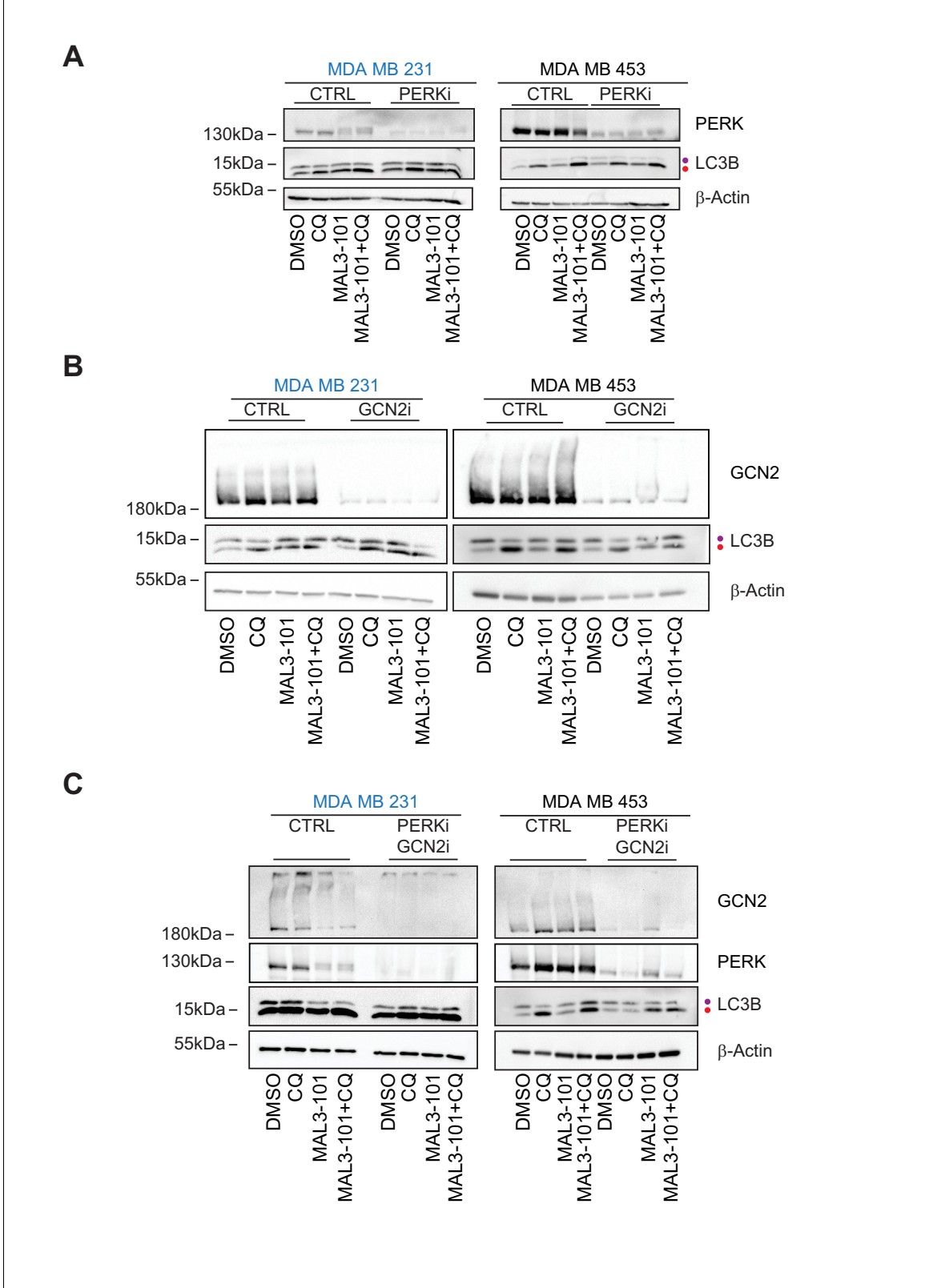

**Figure 12.** GCN2 is required to activate autophagy in MAL3-101-treated resistant cells. (**A**) MAL3-101 sensitive (MDA MB 231, in blue) and resistant (MDA MB 453, in black) cells were transfected with a control siRNA or a mixture of two siRNAs against PERK, and 72 hr after transfection the cells were treated with 12 µM MAL3-101 for 6 hr and CQ was added during the last 2 hr of the treatment to block autophagy. Lysates were prepared for each condition immunoblotted for LC3B to monitor autophagy induction. (**B**) The same cells were used and treated as above except that a control or mixture

*Figure 12 continued on next page*

*Figure 12 continued*

of oligo 1 and 2 against GCN2 was used. (C) The same cells were again used and treated as above except that a control or mixture of siRNAs against both PERK and GCN2 was used. In each panel, β-actin serves as a loading control and the purple and red dot indicate the LC3BI and LC3BII isoforms, respectively.

The online version of this article includes the following source data and figure supplement(s) for figure 12:

**Figure supplement 1.** PERK and GCN2 contribute to autophagy induction in MAL3-101 resistant cells.
**Figure supplement 1—source data 1.** LC3BII and LC3BII/I raw data for *Figure 12—figure supplement 1*.

cells but not in MAL3-101 sensitive lines when Hsp70 activity is perturbed (data not shown). Also in accordance with our discovery, other reports suggested that GCN2/ATF4 signaling plays a role in the survival and proliferation of cell lines and cancer xenografts under suboptimal growth conditions as a result of amino acid/nutrient deprivation and unbalanced protein synthesis (*Harding et al., 2019*; *Nakamura et al., 2018*; *Rajanala et al., 2019*; *Ye et al., 2010*). Moreover, GCN2 sustains mTORC1 suppression when amino acids are limited (*Ye et al., 2015*). Based on our results and these related studies, we conclude that chaperone-mediated proteotoxic stress cannot be attenuated and cancer cell apoptosis is induced when autophagy fails to compensate for Hsp70 inhibition, or when GCN2 activity is impaired.

In summary, we provide evidence that ISR sensors are differentially integrated in proteostasis networks in cancer cells and most likely other cell types. Our work highlights the importance of investigating the interplay between different proteostatic pathways and supports ongoing efforts to modulate GCN2 and eIF2a phosphorylation in cancer (*Kardos et al., 2020*). The targeted regulation of different ISR pathways will also prove valuable in a range of other maladies, including neurodegenerative disease, retinal degeneration, and ischemia (*Chen et al., 2019*; *Grzmil and Hemmings, 2012*; *Halliday et al., 2015*; *Krzyzosiak et al., 2018*; *Luh and Bertolotti, 2020*; *Nguyen et al., 2018*; *Rabouw et al., 2019*; *Zhu et al., 2019*; *Zyryanova et al., 2018*). In parallel, further investigation and chemical optimization of inhibitors of Hsp70 and other chaperones will augment progress on the development of effective combinatorial treatments for breast cancer (*Evans et al., 2010*; *Joshi et al., 2018*).

## Materials and methods

### Cancer cell lines and chemicals

MDA MB 453, MDA MB 468, HCC1419, MCF7, MDA MB 231, HCC1395, BT-474, MDA MB 361, SKBR-3, T47D, HCC38, MDA MB 134, and HCC1937 were purchased from ATCC, and SUM44PE was purchased from Asterand Bioscience Inc (Detroit, MI). Cells were tested to be mycoplasm-free by use of a standard diagnostic PCR using the ATCC universal mycoplasma detection kit. Cell authentication was performed using the University of Arizona Genetics Core according to their protocol. All cell lines were grown at 37°C and 5% $CO_2$. The HCC38, HCC1419, MDA MB 231, MDA MB 453, HCC1395, HCC1937, and MDA MB 468 lines were grown in RPMI-1640 media (#SH30027.FS, GE Healthcare Hyclone, Logan, UT) supplemented with 10% FBS (#SH30071.03, GE Healthcare Hyclone) and 1x penicillin/streptomycin (#15-140-122, Gibco, Thermo Fisher Scientific, Waltham, MA), while BT-474, MDA MB 361, MCF7, T47D cells were grown in DMEM-Hi glucose media with sodium pyruvate (#D6429, Millipore Sigma, Saint Louis, MO) which was supplemented with 10% FBS (GE Healthcare Hyclone) and 1x penicillin/streptomycin (Gibco, Thermo Fisher Scientific, Waltham, MA). SKBR-3 cells were cultured in McCoy's 5A with L-Glutamine media (#SH30200FS, Thermo Fisher Scientific, Waltham, MA) supplemented with 10% FBS (GE Healthcare Hyclone) and 1x penicillin/streptomycin (Gibco, Thermo Fisher Scientific, Waltham, MA). MDA MB 134 cells were grown in L-15 (Leibovitz) media (#L5520, Millipore Sigma, Saint Louis, MO) with 10% FBS (GE Healthcare Hyclone) and 1x penicillin/streptomycin (Gibco, Thermo Fisher Scientific, Waltham, MA), and SUM44PE cells were cultured in DMEM/F-12 media (#11330057, Gibco, Thermo Fisher Scientific, Waltham, MA) supplemented with 2% Charcoal/Dextran stripped FBS (#100–119, Gemini Bio-Products, West Sacramento, CA), along with 5 µg/mL human recombinant insulin (#I2643), 1 µg/mL hydrocortisone solution (#H6909), 5 mM ethanolamine (#E0135), 5 µg/mL human apo-transferrin

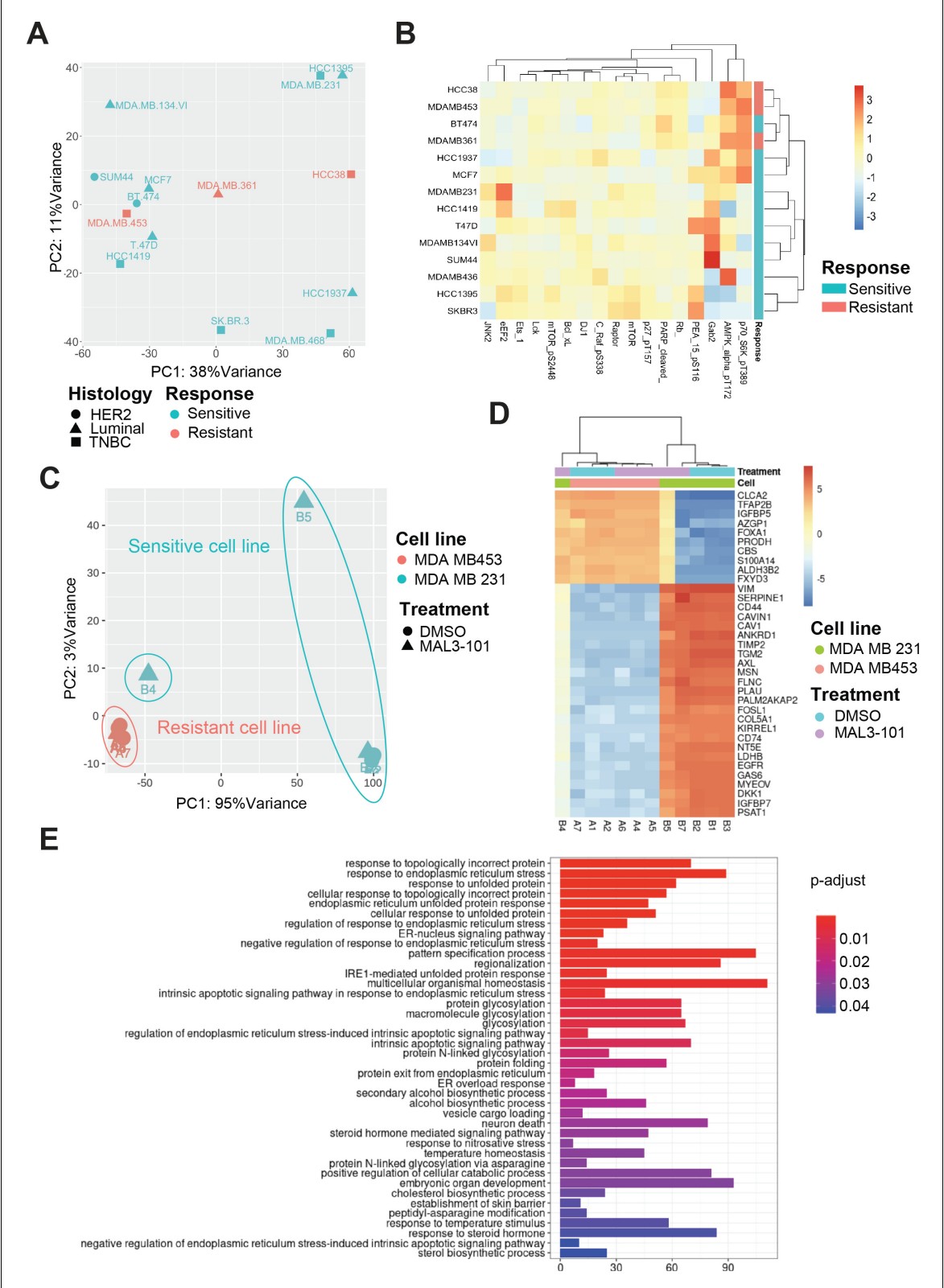

**Figure 13.** Transcriptional upregulation of the unfolded protein response is a signature of MAL3-101 sensitive cells. (**A**) Principal component analysis plot of publicly available RNAseq datasets is presented for tested breast cancer cell lines. MAL3-101-sensitive cells are indicated in aqua green while resistant cell lines are indicated in light red. (**B**) Heatmap representation of significantly differentially expressed proteins between publicly available RPPA datasets for sensitive and resistant cells (MDA MB 231 and MDA MB 453). Wilcoxon signed-rank test between resistant and sensitive cell lines was

*Figure 13 continued on next page*

*Figure 13 continued*

used to determine proteins that were differentially expressed. p<0.1 was considered significant. (C) Principal component analysis plot of RNAseq performed on MAL3-101 treated (Samples A4, **A5, A6, B4, B5, B7**) or DMSO (Samples A1, **A2, A7, B1, B2, B3**) treated sensitive and resistant cells. MDA MB 453 cells (MAL3-101 resistant, sample A) are represented in light red and MDA MB 231 cells (MAL3-101 sensitive, sample B) in aqua green. (D) The top 35 differentially expressed genes detected in MAL3-101 treated and untreated cell lines (MDA MB 231 in green and MDA MB 453 in light red) are represented in a heatmap graph. (E) Gene Ontology enrichment plot of Biological Processes using significantly differential genes in MDA MB 231 cell lines with and without MAL3-101 treatment. p<0.05 was considered significant.

(#T2252), 6.6 ng/mL 3′,3′,5-triiodo-ʟ-thyronine sodium salt (#T5518), and 8.7 ng/mL sodium selenite (#S9133), all purchased from Millipore Sigma, Saint Louis, MO.

MAL3-101 (*Fewell et al., 2004*), MG 132 (EMD Millipore, Burlington, MA), everolimus (Sigma-Aldrich, Saint Louis, MO), 17-AAG, and CB-5082 (Selleck Chemicals, Houston, TX) were dissolved in DMSO to a final concentration of 20 mM, while bafilomycin (Selleck Chemicals, Houston, TX) and bortezomib (Cell Signaling Technology, Danvers, MA) were prepared at a final concentration of 1 mM in DMSO. ISRIB and tunicamycin (Sigma-Aldrich, Saint Louis, MO) were suspended to a final concentration of 10 mg/mL and GSK-2606414 (GSK; Tocris Bioscience, Bristol, UK) was dissolved to a final concentration of 50 mM in DMSO. Chloroquine (CQ; Sigma-Aldrich, Saint Louis, MO) was dissolved in sterile water to a final concentration of 100 mM. All compounds were stored at −20℃. Prior to use, the desired amount of each compound was added to pre-warmed media, mixed thoroughly, and added directly onto cells.

## siRNA transfection

To silence ATG5, cells were seeded at a density of 150,000 in a six-well plate, and after 24 hr, two different siRNA oligos against ATG5 (siGENOME9474, D-004374–03 and D-004374–05, Dharmacon, Lafayette, CO) or a control siRNA (BLOCK-iT Alexa Fluor Red Fluorescent Control, Invitrogen, Thermo Fisher Scientific) were transfected using Lipofectamine RNAiMAX (Invitrogen, Thermo Fisher Scientific) at 20 nM according to the manufacturer's instructions. After 6 hr, cells were harvested and seeded at a density of 1,500 cells/100 µL in 96-well clear-bottomed plates (Greiner bio-one, North Carolina) for cell viability assays (see below). The remaining transfected cells were collected for protein extraction 72 hr post-transfection to detect knock-down efficiency, as described below. The optimal time frame and concentration of siRNA were selected after conducting time course experiments in which ATG5-12 and Hsp70 protein abundance were analyzed by immunoblot (data not shown).

To silence specific ISR sensors, MDA MB231 and MDA MB 453 cells were seeded at a density of 1,000,000 in a 10 cm dish, and after 24 hr two oligos for PERK (EIF2AK3), PKR (EIF2AK2) (#s18101 and #s18103, #s11186 and #s11185, Ambion Fisher Scientific Waltham, MA) or GCN2 (EIF2AK4) (#D-005314–16, #D-005314–13, Dharmacon, Lafayette, CO), or one oligo for HRI (EIF2AK1) (#D-005007–04) or a control siRNA (BLOCK-iT Alexa Fluor Red Fluorescent Control, Invitrogen, Thermo Fisher Scientific) were transfected using Lipofectamine RNAiMAX (Invitrogen, Thermo Fisher Scientific) at 20 nM according to the manufacturer's instructions. After 48 hr, cells were harvested and seeded at a density of 150,000 cells/well in a six-well plate. After 24 hr, cells were treated with the indicated compounds and processed for protein extraction, as described below. In the experiment where both GCN2 and PERK were knocked down 5 nM of each siRNA oligo (#s18101 and #s18103, #D-005314–16 and D-005314–13) where transfected using Lipofectamine RNAiMAX and the samples where processed as indicated before in the single knock down experiments.

For experiments in which the UPR transducer Ire1α was knocked down, MDA MB 231 and MDA MB 453 cells were seeded at a density of 1,200,000 in a 10 cm dish, and after 24 hr, a set of oligos at 10 nM for Ire1α (ERN1) (#s200430, Ambion Fisher Scientific, Waltham, MA) or a control siRNA were transfected as above. After 24 hr, cells were harvested and seeded at a density of 150,000 cells/well in a six-well plate, and after 24 hr, cells were treated with the indicated compounds and processed for protein or RNA extraction (see below).

## Generation of stable cell lines and confocal microscopy

To transfect the plasmid encoding the RFP-GFP-LC3B tandem autophagy reporter (provided by Dr. Charlene Chu, Department of Pathology, University of Pittsburgh) (*Kimura et al., 2007*), MDA MB

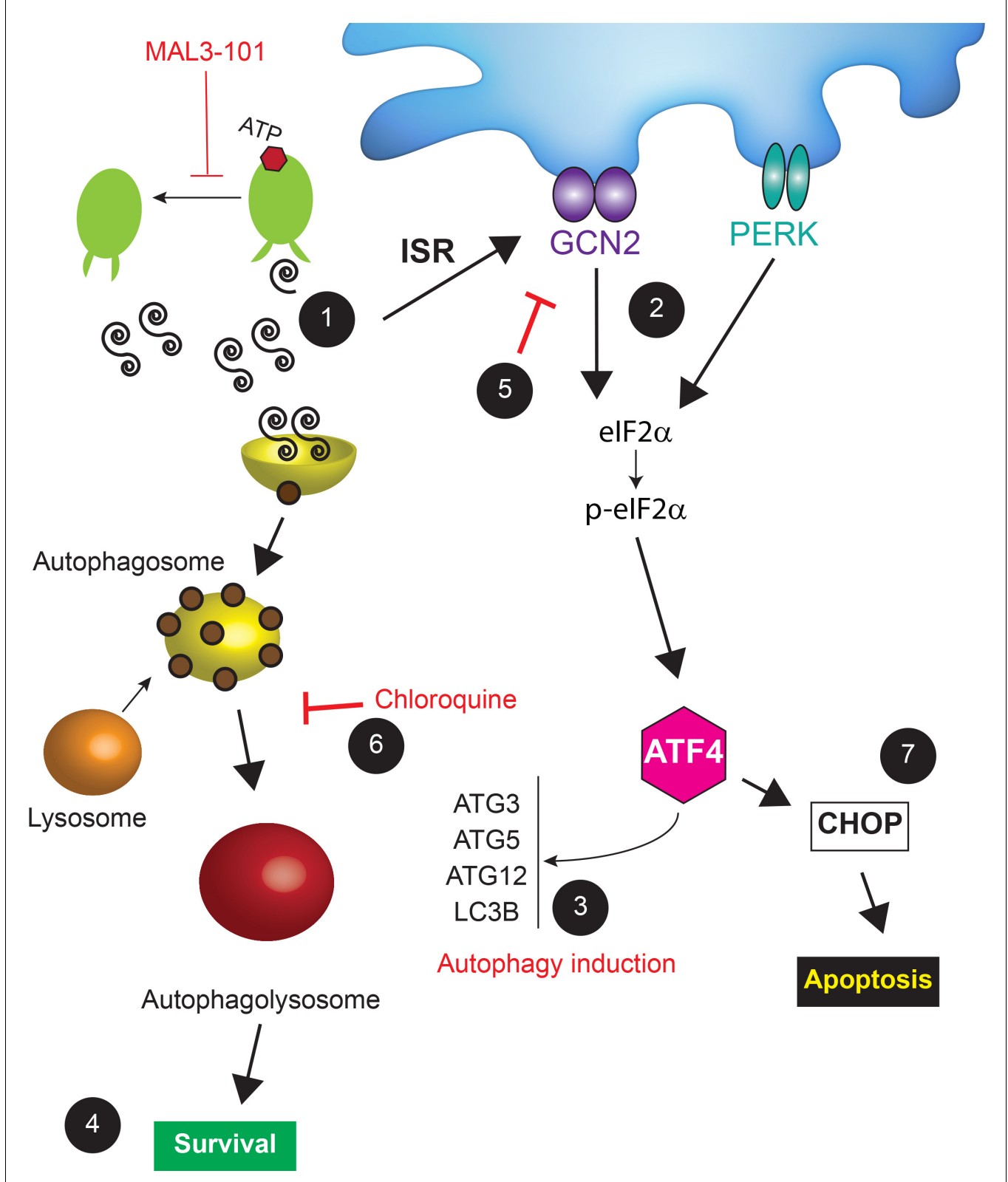

**Figure 14.** A model for cancer cell adaptation to Hsp70 inhibition. (1) In MAL3-101 resistant and sensitive cells, Hsp70 inhibition lead to activation of the ISR due to the accumulation of misfolded proteins (only misfolded proteins in the cytoplasm are shown). (2) GCN2-mediated eIF2α phosphorylation, which can be induced by amino acid starvation (see Discussion), also induces (3) the transcription of autophagy-related genes, favoring autophagy induction and (4) protecting MAL3-101-resistant breast cancer cells. (5) If GCN2 activity is impaired or (6) autophagy is inhibited (for *Figure 14 continued on next page*

*Figure 14 continued*

example by chloroquine), Hsp70 inhibition is cytotoxic so (7) CHOP accumulates and resistant cells undergo apoptosis. (7) In contrast, in MAL3-101-sensitive cells, PERK induces ATF4 and CHOP, which results in cell death.

231 and MDA MB 453 cells were seeded at a density of 1,200,000 in 10 cm dish, and after 24 hr, a total of 12 µg of the plasmid was introduced using the FuGENE6 reagent (Promega, Madison, WI) according to the manufacturer's instructions. Cells were selected for G418 resistance (350 µg/mL, GE Healthcare Hyclone) for 8 days, and fresh G418-containing media was added every 3 days. Once a G418 resistant polyclonal population was obtained, the cells were trypsinized and resuspended at a density of 1 cell/100 µL in a 96-well plate. After 2 weeks, single clones where isolated, expanded, and tested for RFP-GFP-LC3B reporter expression by microscopy. MAL3-101 sensitivity in the obtained clones was tested as described in assays for cell viability (see below; two clones for MDA MB 231 and MDA MB 453 lines, data not shown). One clone for each line was used to test the autophagy induction rate upon Hsp70 inhibition in MAL3-101. Each RFP-GFP-LC3B expressing clone was seeded at a density of 200,000 cells in poly-D-Lysine Coated 35 mm MatTek dishes (P35GC-1.5–10 C, MatTek Corporation, Ashland, MA). After 24 hr, cells were independently treated with EBSS, MAL3-101 (5 µM), or vehicle (DMSO) for 22 hr. The samples were observed with a Nikon A1 point scanning confocal microscope without fixation to measure autophagosome and autophagolysosome formation by monitoring the distribution and alteration of RFP and GFP fluorescent signal after the indicated treatments. Complete volumes of cells were acquired at 0.2 µm steps, and volumes were reconstructed and analyzed using Nikon's NIS-Elements software (Nikon Instrument, Melville, NY). Bright spot detection tool was used to identify and quantify the number of RFP and GFP positive ('dots') per cell after equalizing signal intensity suing the LUTs - Non-destructive Image Enhancement tool. A non-parametric Kruskal-Wallis test analysis was performed using Prism software (GraphPad, La Jolla, CA). Asterisks in figures indicate statistically significant differences between the samples. To obtain representative images from this experiment, maximum intensity projections of 0.2 µm steps though the entire cell were generated using Nikon's NIS-Elements software. The scale shown is 10 µm.

## Assays for cell viability

Cells were plated at the indicated densities in *Table 2* in 96-well clear-bottomed plates (Greiner bio-one, North Carolina), and after 72 hr, of the indicated treatment the cells were lysed and incubated with the CellTiter-Glo reagent (Promega) and luminescence was read on a Bio-Rad ChemiDoc XRS+ with the associated Image Lab software (Bio-Rad, Thermo Fisher Scientific). Under conditions in which cells were treated with more than one compound (e.g. *Figure 3A and B*; *Figure 3—figure supplement 1A*; *Table 3*), the cells were incubated with a range of doses of MAL3-101 in the presence of subcritical doses of bortezomib, everolimus, CQ, or bafilomycin that induced no greater than 30% of cell death in each line 72 hr after treatment at these concentrations (see *Table 2*). Viability was quantified after normalizing values obtained after bortezomib, CQ, or bafilomycin treatment alone at the indicated concentrations (*Tables 2* and *3*). When viability was measured after ATG5 knock-down, MAL3-101 sensitivity was measured 48 hr after treatment and 72 hr after siRNA transfection.

Analysis of cell surface Annexin-V was performed by staining cells with the Annexin-V Apoptosis Detection Kit (eBioscience, Thermo Fisher Scientific) following the manufacturer's instructions. In brief, cells were seeded at a density of 220,000 cells/well for the MDA MB 361 line and at 150,000 cells/well for the other indicated lines in 6-well plates and allowed to adhere overnight before performing the indicated treatments. After mild trypsinization, the cells were harvested and washed in ice-cold PBS and then equilibrated in Annexin-V binding buffer. A cell resuspension of 150,000 cells in 100 µL was incubated with 5 µL of Annexin-V APC for 15 min in the dark. After three washes with binding buffer, the cells were suspended in 500 µL of Annexin-V binding buffer containing 5 µL of Propidium iodide (PI) and analyzed on an Accuri C6 FACS apparatus (BD-Biosciences, San Diego). Single-cell staining was used to discriminate Annexin-V- and PI-positive cells, respectively. Annexin-V and Annexin-V/PI double positive cells were summed and are represented as the percentage of apoptotic cells.

## Immunoblot analysis

All antibodies used in this study were obtained from Cell Signaling Technology (Danvers, MA), unless indicated otherwise. To measure the levels of endogenous proteins, cells were plated at a density of 150,000 cells/well for all cell lines except for MDA MB 361 cells (220,000 cells/well) in six-well plates and allowed to adhere overnight before the indicated treatments. The supernatant/media was collected, and after the cells were washed in PBS cell lysates were obtained after resuspending adherent cells in 1% SDS RIPA buffer (100 mM Tris-HCl, pH 7.5, 1% NP40, 1% SDS, 300 mM NaCl and 0.5% sodium deoxycholate) supplemented with protease inhibitors (cOmplete Mini EDTA free tablets; Roche, Indianapolis, IN), phosphatase inhibitors (PhosSTOP, phosphatase inhibitor tablets; Millipore Sigma, Saint Louis, MO), 10 mM N-ethylmaleimide (NEM; Sigma-Aldrich), and 5 mM phenylmethanesulfonyl fluoride (PMSF; Sigma-Aldrich) on ice. The lysate was clarified by sonication and centrifugation at 13,000 g for 5 min at 4°C, and total protein was quantified with the BCA assay kit (Thermo Fisher Scientific). Aliquots containing 35 μg of total protein were incubated in sample buffer at 95°C for 5 min, subjected to SDS-PAGE using a 15% polyacrylamide gel, and after transfer to PVDF membranes (Immobilon-FL, #IPFL85E, Sigma-Aldrich) the blots were incubated with the following antibodies: anti-LC3B (2775S; at 1:1000), anti-ATG5 (D5F5U, #12994S; at 1:1000), anti-p62/SQSTM1 (P0067, Sigma-Aldrich; at 1:2000), anti-CHOP (L63FZ, #2895S; at 1:500), anti-cleaved Caspase-3 (#9661S; at 1:750), and anti-cleaved Caspase-7 (D6H1, #8438S; at 1:500). For the remaining proteins aliquots from the same lysates were instead heated to 37°C for 30 min prior to SDS-PAGE using 4–20% Tris-Glycine gradient gels (XP04202Box, Thermo Fisher Scientific), and after transfer on nitrocellulose membranes (#84–874, Prometheus Laboratories Inc, California), the blots were incubated with anti-cleaved Caspase-8 (18C8, #9496; at 1:1000), anti-BiP (C50B12, #3177S; at 1:1000), anti-Hsp70 (smc-113, StressMarq Biosciences, Victoria, British Columbia; at 1:1000), anti-Hsc/Hsp70 (#4872S; at 1:1000), anti-HER2 (29D8, #2165; at 1:2000), anti-eIF2α (#9722S; at 1:2000), anti-phospho-eIF2α (#9721L; at 1:500), anti-PERK (D11A8, #5683S; at 1:2000), anti-PKR (D7F7, #12297S; at 1:2000), anti-Ire1α (14C10, #3294S; at 1:1000), anti-HRI (MBS2538114, MyBioSource, San Diego, CA; at 1:500) and anti-GCN2 and anti-GCN2 pT899 antibody (ab-134053 and ab-75836, Abcam, Cambtidge, UK; at 1:2000). Anti-β-actin (ab-6276, Abcam, Cambtidge, UK; at 1:5000) was used as loading control. All primary antibodies were incubated overnight at 4°C. Anti-rabbit immunoglobulin G (IgG) and anti-Mouse immunoglobulin HRP-conjugated secondary antibodies (at 1:4000) were applied for 2 hr at room temperature prior to imaging. In all cases, proteins were visualized using a mix of ProSignal Pico and ProSignal Femto ECL Reagent at a 1:1 ratio (Prometheus Laboratories Inc, California) and images were taken using a Bio-Rad ChemiDoc XRS+ with Image Lab software. Data were analyzed using ImageJ software (National Institutes of Health, NIH).

To measure LC3BII accumulation and LC3BI to LC3BII conversion in the presence or absence of MAL3-101, cells were seeded as above in six-well plates, and after 24 hr, the cells were treated with 12 μM MAL3-101 or with DMSO for 6 hr. A total of 2 hr before the end of the 6 hr time point, 250 μM CQ was added for the remaining 2 hr. A corresponding volume of vehicle (i.e. water) was added for the CQ mock treated samples. Cells were processed and LC3B abundance was detected as described above. The CQ concentration was selected after conducting experiments with increasing doses (50, 100, 150, 200, 250, 300, and 350 μM) in which LC3BII accumulation was analyzed by immunoblot and apoptosis induction was monitored by cleaved caspase-3 and caspase-7 appearance (data not shown). The CQ concentration, which gave the highest LC3BII accumulation but no significant apoptosis induction after a 2 hr treatment, was selected.

To induce non-selective (macro) autophagy, cells were washed twice in sterile PBS to remove any remaining media and then starved for 6 hr in Earle's Balanced Salt Solution (EBSS, Thermo Fisher Scientific). A total of 2 hr before the end of the 6 hr time point, 250 μM CQ was added for 2 hr. A corresponding volume of vehicle (i.e. water) was added for the mock treated samples.

To inhibit PERK, cells were pre-treated with GSK-2606414 at a final concentration of 2 μM for 2 hr in complete media and then 12 μM MAL3-101 or a corresponding volume of vehicle (DMSO) was added for 6 hr. To inhibit the ISR pathway, resistant and sensitive cells were treated with 10 μM ISRIB for for 10 hr and 15 μM MAL3-101 or a corresponding volume of DMSO was added for 6 hr before the 10 hr endpoint. Cells were processed as described above.

To detect levels of total protein ubiquitination, cells were seeded at the density mentioned above in six-well plates and allowed to adhere overnight, before they were treating with 18 nM bortezomib

for 4 hr. Samples were collected and processed as above and 10 μg of total protein was subject to SDS-PAGE and proteins were transferred to nitrocellulose. Prior to incubating the blots in a milk solution, the nitrocellulose membranes were incubated in a boiling water bath for 1 hr to expose antibody epitopes. An anti-ubiquitin antibody, P4D1 (Santa Cruz Biotechnology, Dallas, TX), was used at 1:1000, and anti-mouse immunoglobulin G (IgG) HRP-conjugated secondary antibodies (at 1:4000) were applied for 2 hr at room temperature prior to imaging.

To detect ATF6, cells were plated at a density of 220,000 cells/well for the MDA MB 361 cells and 150,000 cells/well for the other cell lines in six-well plates and allowed to adhere overnight. Next, the cells were treated as indicated and 2 mM DTT was applied for 1 hr as a positive control. All treatments were performed in the presence of 5 μM MG 132 to avoid ATF6 degradation (*Horimoto et al., 2013*). The supernatant/medium was then collected, and after the cells were washed in PBS containing 5 μM MG 132, the cells were detached with trypsin, centrifuged in a clinical centrifuge, and the cell pellets were resuspended in 10 mM Tris-HCl, pH 7.5, 150 mM NaCl, 10 μM MG 132, and 1% SDS buffer, supplemented with protease inhibitors (cOmplete Mini EDTA free tablets; Roche, Indianapolis, IN), 10 mM NEM (Sigma-Aldrich), and 5 mM PMSF (Sigma-Aldrich). The lysate was then clarified by sonication and centrifugation at 13,000 g for 5 min at 4°C, and total protein was quantified with the BCA assay kit as described above. A total of 50 μg protein was incubated at 95°C for 5 min, subjected to SDS-PAGE using a 8% polyacrylamide gel (acrylamide:Bis-acrylamide, 29:1, 40% solution, OmniPur, Sigma-Millipore, Germany), and the resulting nitrocellulose filter was incubated in Tris-buffered saline with 0.1% Tween 20 (TBST) containing 10% non-fat dry milk for 1 hr at room temperature with gentle shaking. The nitrocellulose filter was then incubated with an aliquot of anti-ATF6 antibody (73–500, Bio Academia, Japan; at 1:1000) in the presence of 5% non-fat dried milk powder for 1 hr at room temperature. An anti-light chain-specific anti-mouse IgG monoclonal antibody (115-035-174, Jackson ImmunoResearch, West Grove, PA) was then used as the HRP-conjugated secondary antibody at 1:5000 in 1% non-fat milk TBST for 2 hr at room temperature. Decorated proteins were visualized using the ProSignal Femto ECL Reagent kit (Prometheus Laboratories Inc, California) and images were taken using a Bio-Rad ChemiDoc XRS+ with Image Lab software. Full-length ATF6 was quantified as an indicator of ATF6-mediated UPR induction using ImageJ software.

## Real-time quantitative PCR

Cells were seeded at 150,000 cells/well for MDA MB 231, MDA MB 453, and MCF7 cells and at 220,000 cells/well for the MDA MB 361 line in six-well plates, grown overnight, and then treated for the indicated times with the indicated compounds. RNA was extracted using the RNeasy kit (Qiagen, Hilden, Germany) according to the manufacturer's instructions. cDNA was synthetized from 1 μg of the extracted RNA using qScript cDNA SuperMix reverse transcriptase kit (Quantabio, Beverly, MA), and 70 ng were used for Real Time qPCR using the QuantStudio 3 (Thermo Fisher Scientific). Each PCR was performed on three or more biological replicates and with three technical replicates for each reaction. Primer efficiency was determined by serial dilution of the template and the relative expression ratios were calculated (*Pfaffl, 2001*). Primers amplifying β-actin were used as an internal control. All primer sequences were published in previous work (*Sabnis et al., 2016*; *Sannino et al., 2018*).

## Proteasome activity assays

Cell lysates from MDA MB 231, MDA MB 453, MDA MB 361, and MCF7 cells were collected as described (*Milan et al., 2015*), and proteasome activity was assessed by monitoring the production of 7-amino-4-methylcoumarin (AMC) from the Suc-LLVY-AMC proteasome substrate (cat. no. I-1395; Bachem, Torrance, CA), which specifically detects the chymotrypsin-like activity of the proteasome (*Stein et al., 1996*). To this end, 10 μL aliquots of each lysate were incubated with 1 μL of a 5 mM stock solution of Suc-LLVY-AMC in 20 mM Tris-HCl, pH 7.5, 2 mM ATP, 2 mM $MgCl_2$, and 0.2% bovine serum albumin in the presence or absence of 10 μM MG 132 (*Gleixner et al., 2017*; *Sannino et al., 2018*). The fluorescence of released AMC was measured at time 0 and after 30 min in a Cytation5 plate reader at an excitation wavelength of 380 nm and an emission wavelength of 460 nm (BioTek, Winooski, Vermont). To calibrate the assay, a standard free fluorophore solution containing a range of AMC concentrations was used (VWR, Randor, PA). All measurements were

performed in duplicate and values were normalized to protein content as determined by the BCA protein assay kit according to the manufacturer's instructions (Thermo Fisher Scientific). Proteasome activity was calculated by: ((RFU2 – RFU1) / (T2-T1)) / [protein]. One unit of proteasome activity is defined as the amount of proteasome that generates 1.0 picomol of AMC per min at 37°C.

## RNA sequencing and sequence analysis

MDA MB 231 and MDA MB 453 cells were seeded at 150,000 cells/well in six-well plates, grown overnight, and then treated for 6 hr with a final concentration of 12 µM MAL3-101 or an equal volume of DMSO. RNA was extracted using the RNeasy kit (Qiagen, Hilden, Germany) according to the manufacturer's instructions. Each total RNA extraction was performed on three biological replicates. RNA sequencing library construction, quality control, and PE150 sequencing were performed by Novogene Co., Ltd. according to their protocol. Transcript counts from all samples were quantified with Salmon v. 1.3.0 (*Patro et al., 2017*), converted to gene-level counts with tximport (*Soneson et al., 2015*), and were then used in the DESeq2 function: 'DESeqDataSetFromTximport' (*Love et al., 2014*).

Differential gene expression analysis was performed using the DESeq2 package (*Love et al., 2014*). Principal component analysis plots were generated from variance stabilizing transformed (VST) DESeq2 data-frames. For RNAseq datasets corresponding to MAL3-101 treated and untreated cells, the top 35 VST differentially expressing genes were displayed in *Figure 13D*. Differentially expressed genes with a p-value below 0.05 were used for Gene Ontology analysis applying the 'clusterProfiler' package (*Yu et al., 2012*; *Figure 13E*). The RNAseq data is deposited onto the Gene Expression Omnibus database (GEO, GSE178352). All raw data are available upon request from the corresponding authors.

## Gene and protein expression cluster analysis

For publicly available RNAseq datasets, RNAseq fastq files from the 14 breast cancer cell lines tested for MAL3-101 sensitivity in *Figure 1A* were downloaded with GEO Accession Numbers GSE36133 (CCLE) or GSE73526 (*Barretina et al., 2012*; *Marcotte et al., 2016*). Transcript counts from all samples were quantified with Salmon v.0.12.0 (*Patro et al., 2017*) and converted to gene-level counts with tximport (*Soneson et al., 2015*). Differential gene expression analysis was performed using the DESeq2 package (*Love et al., 2014*). Publicly available RPPA data was obtained from the Neel Lab (http://neellab.github.io/simem/) (*Marcotte et al., 2016*) and normalized $Log_2$ median centered reads were used for analysis. Data visualization was performed using 'ggpubr' and the CRAN 'heatmap' packages.

## Statistical analysis

$IC_{50}$ concentrations from CellTiter-Glo assays were calculated as described (*Sabnis et al., 2016*) using SigmaPlot 11.0 (Systat Software, Inc). GraphPad Prism (GraphPad Software, Inc) was used to carry out a two-tailed Student's t test between two samples and a non-parametric Kruskal-Wallis test analysis was performed to establish statistical significance between control and compound-treated samples in the same line for the RFP-GFP-LC3B autophagy reporter assay (*Figure 5*). In all experiments, $p<0.05$ was considered significant with the exception of the data presented in *Figure 13B*, where protein expression levels between resistant and sensitive cell lines with a p-value below 0.1 were displayed.

## Acknowledgements

We thank Dr. Tia-Lynn Ashman for reagents and instrumentation used in this study. We also thank Dr. Simon Watkins and the CBI imaging center for access to their workstation data analysis facility, Dr. Alexandra Manos-Turvey for the preparation of MAL3-101, Dr. Ronald C Wek for providing the plasmid for GCN2 overexpression, and Ms. Taber S Maskrey for LCMS QC of MAL3-101 batches. This work was supported by grants from the long-term European Molecular Biology Organization (EMBO) via a post-doctoral fellowship (ALTF 823–2016) to SS, from the National Institutes of Health (F30CA250167) to MEY, and from the National Institutes of Health (GM131732 and DK79307), the University of Pittsburgh/UPMC Institute for Precision Medicine, the UPMC Hillman Cancer Center supported by National Institutes of Health award number P30CA047904, Pilot Funding from the

Translational and Precision Pharmacology programs, and by a Howard Hughes Medical Institute Collaborative Innovation award to JLB AVL and SO are Hillman Fellows. Screening was supported by Cancer Center Support Grant (CCSG), P30 CA047904. We also thank David Zaidins for technical assistance.

## Additional information

### Funding

| Funder | Grant reference number | Author |
| --- | --- | --- |
| European Molecular Biology Organization | ALTF 823-2016 post-doctoral fellowship | Sara Sannino |
| National Institutes of Health | F30CA250167 | Megan E Yates |
| National Institutes of Health | GM131732 | Jeffrey L Brodsky |
| National Institutes of Health | DK079307 | Jeffrey L Brodsky |
| National Institutes of Health | P30CA047904 | Jeffrey L Brodsky |
| Howard Hughes Medical Institute | Howard Hughes Medical Institute Collaborative Innovation award | Jeffrey L Brodsky |
| University of Pittsburgh | Translational and Precision Pharmacology programs (pilot grant) | Jeffrey L Brodsky |

The funders had no role in study design, data collection and interpretation, or the decision to submit the work for publication.

### Author contributions

Sara Sannino, Conceptualization, Resources, Data curation, Formal analysis, Funding acquisition, Validation, Investigation, Visualization, Methodology, Writing - original draft; Megan E Yates, Data curation, Software, Formal analysis, Validation, Investigation, Methodology; Mark E Schurdak, Data curation, Formal analysis, Supervision, Validation, Investigation, Writing - review and editing; Steffi Oesterreich, Resources, Supervision, Writing - review and editing; Adrian V Lee, Resources, Supervision, Methodology, Writing - review and editing; Peter Wipf, Formal analysis, Supervision, Validation, Project administration, Writing - review and editing; Jeffrey L Brodsky, Conceptualization, Resources, Supervision, Funding acquisition, Validation, Methodology, Project administration, Writing - review and editing

### Author ORCIDs

Sara Sannino (iD) https://orcid.org/0000-0002-9125-9098
Jeffrey L Brodsky (iD) https://orcid.org/0000-0002-6984-8486

### Decision letter and Author response

Decision letter https://doi.org/10.7554/eLife.64977.sa1
Author response https://doi.org/10.7554/eLife.64977.sa2

## Additional files

### Supplementary files

• Transparent reporting form

### Data availability

All data generated or analysed during this study are included in the manuscript and source files.

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
