## [Decision Letter]

**Acceptance summary:**

This study systematically explored the response of breast cancer cells to proteotoxic challenges induced by inhibition of the chaperone Hsp70. Interestingly, resistance to Hsp70 inhibition correlated with high levels of autophagy and required the integrated stress response transducer GCN2 that is typically associated with amino acid starvation. In contrast, sensitivity to Hsp70 inhibition was mediated by activation of stress signaling via PERK. Based on these results, future studies can be designed to provide further insight into how cancer cells achieve resistance to proteotoxic stress.

**Decision letter after peer review:**

Thank you for submitting your article "Unique integrated stress response sensors regulate cancer cell susceptibility when Hsp70 activity is compromised" for consideration by *eLife*. Your article has been reviewed by 3 peer reviewers, and the evaluation has been overseen by a Reviewing Editor and Maureen Murphy as the Senior Editor. The reviewers have opted to remain anonymous.

The referees find your work interesting and recommend publication after appropriate revisions. All three reviewers agreed that additional mechanistic studies would significantly improve the manuscript (and make it acceptable for publication).

While the authors have performed numerous experiments to define the protein homeostasis remodeling associated with MAL3-101 sensitivity in breast cancer cell lines, further experiments are necessary regarding the conclusion that sensitive and resistant cells are differentially dependent on GCN2 and PERK for dictating survival or apoptosis. Although the idea that GCN2 and PERK (or other ISR kinases) can initiate distinct signaling in response to different types of insults is not new, this work does organize this idea in a defined way and adds new results that can continue to inform that area of study. The key question is how does ISR signaling through PERK and GCN2 integrate with other functional inputs to dictate distinct cellular responses. This is likely achieved through mechanisms including the posttranslational modification of different downstream transcriptional regulators, integration of ISR signaling with other stress-responsive signaling pathways, or through heterooligomerization of ISR transcription factors with other inputs (e.g., heterooligomerization of ATF4 with other b-ZIP transcription factors). The authors are in a unique position to address these questions, but ultimately, they are not pursued in this manuscript.

Essential revisions:

1. One interesting question emerging from this study is: What are the other inputs for resistant and sensitive cells that differentially alter ISR signaling outputs in these two models? This could be addressed by performing RNAseq of resistant or sensitive cells treated with Mal3-101 with or without depletion of Gcn2 or Perk. This would provide a better framework to interpret the differential ISR signaling between these two types of cell lines and likely reveal other signaling mechanism that serve as modulators of ISR signaling.

2. There are publicly available transcriptional profiles of many of these breast cancer cell lines, so a clustering analysis might reveal the patterns of transcription that define the sensitivity groups, shedding light on potential mechanisms in an unbiased way. Does sensitivity correlate to known driver oncogenes? Drug efflux protein expression? Amino acid transporters? Metabolism? Translation control machinery? More colloquially – the classical breast cancer categorization subtype (e.g. TNBC, luminal, HER2) do not correlate with Hsp70 inhibitor sensitivity, so what does at a molecular level? Similarly, the authors also systematically examined resistance to the Hsp90 inhibitor, 17-AAG, so a parallel analysis of transcriptional profiles could provide further insight into the similarities and differences.

3. Is GCN2 activation necessary/sufficient for Hsp70 inhibitor resistance. Does over-expression of GCN2 drive resistance in an otherwise sensitive cell type? How about amino acid starvation? mTor inhibitor? It seems likely that the story could be more complex, but a specific test would be informative either way.

Other comments:

1. The MAL3-101 compound has been reported previously, but considering the way it is used, it would be good to have a statement up front in the text about its selectivity and known engagement of different HSP70s. Which ones it hits, EC50, etc. For example, the potential for this compound to target BiP?. What about mitochondrial HSP70? This is addressed in the results and discussion, but it would be good to include this more clearly in the intro.

2. The differential activation of PERK and GCN2 in sensitive and resistant cells should be demonstrated more directly. For example, a phosphor-PERK and phosphor-GCN2 assay to show increased activity of these two eIF2a kinases should be done.

3) Overall, the paper appears too long and results seem a bit scattered across the many chapters/figures. Combining some data to focus on key messages would be very helpful.

4) Line 349: Please correct HSPSA5 to HSPA5.

5) Please correct protein/gene nomenclature for clarity (capitalized for human, e.g. lines 352-361).

6) The increase in ubiquitinated proteins upon proteasome inhibition may still be proteasome dependent. The capacity of the proteasome is sufficient for the higher loads and differences could be explained by e.g. increased E3-Ligase activity. Particularly in the light of the finding that bortezomib sensitized some of the resistant cell lines to MAL3-101 treatment as well (although to a lesser extent than CQ), it would be helpful to further discuss possible explanations.

*Reviewer #1:*

Here, the authors use a combined chemical and genetic approach to probe protein homeostasis remodeling in breast cancer cell lines treated with the HSP70 inhibitor Mal3-101. They identify select breast cancer cell lines of diverse types (e.g. luminal, TNBC) that are differentially sensitive to treatment with MAL3-101. Using RT-qPCR and immunoblotting, the authors show that this differential sensitivity does not appear to be associated with remodeling of HSP70 chaperoning activity. So the authors used pharmacologic and genetic strategies to define the underlying basis for the differential sensitivities to this HSP70 inhibitor. Initially, the authors used inhibitors of proteasome- and autophagy-associated degradation to demonstrate that breast cancer cell lines resistant to Mal3-101 have increased dependence on protein degradation. However, only pharmacologic inhibition of autophagy restored resistance of these lines to Mal3-101. These results and others probing autophagic flux in resistant cell lines treated with Mal3-101 indicate these cells increase autophagy to improve survival in response to pharmacologic HSP70 inhibition. The authors then probe the dependence on this increased autophagic flux on UPR and ISR signaling using both sensitive and resistant breast cancer cell lines. Through these efforts, the authors show that genetic depletion of the UPR/ISR kinase PERK selectively blocks MAL3-101-induced apoptosis in sensitive cells, while having no significant effect on resistant cells. Alternatively, genetic depletion of the ISR kinase GCN2 blocked increased autophagic flux and increased apoptosis in resistant cells, while minimally impacting sensitive cell lines. These results indicate that sensitive and resistant cell lines differentially rely on the PERK and GCN2 to dictating apoptosis or survival.

Overall, the authors have clearly performed numerous experiments to define the protein homeostasis remodeling associated with Mal3-101 sensitivity in breast cancer cell lines. Further, their concluding results that sensitive and resistant cells are differentially dependent on GCN2 and PERK for dictating survival or apoptosis, respectively are interesting. However, the idea that GCN2 and PERK (or other ISR kinases) can initiate distinct signaling in response to different types of insults is not exactly a new idea in the field, although admittedly, this work does organize this idea in a defined way and adds new results that can continue to inform that area of study. The key question is how does ISR signaling through PERK and GCN2 integrate with other functional inputs to dictate distinct cellular responses. This is likely achieved through mechanism including the posttranslational modification of different downstream transcriptional regulators, integration of ISR signaling with other stress-responsive signaling pathways, or through heterooligomerization of ISR transcription factors with other inputs (e.g., heterooligomerization of ATF4 with other b-ZIP transcription factors). The authors work uniquely positions them to address these types of questions, but ultimately, they are not pursued in this manuscript. In the current form, the manuscript sets up the next series of very interesting studies to get the bottom of this type of differential regulation. To be fair, it is not reasonable to expect the entire underlying mechanism highlighted by this work to be defined in a single paper, but I do think it is important to address some aspects of the differential sensitivity from either a biological or therapeutic perspective.

Below I include a few suggested experiments to bring some more closure to the manuscript. All are not necessary, especially considering the amount of work already included, but one of these types of experiments would significantly improve the impact of the work. There are, of course, other potential experiments too that would be suitable to bring closure to this story, but I think something similar to the below is required.

1. One interesting question emerging from this study is: 'what are the other inputs for resistant and sensitive cells that differentially alter ISR signaling outputs in these two models?'. This could potentially be addressed by performing RNAseq of resistant or sensitive cells treated with MAL3-101 and depleted of either GCN2 or PERK. This would provide a better framework to interpret the differential ISR signaling between these two types of cell lines and likely reveal other signaling mechanism that serve as modulators of ISR signaling.

2. Similar to #1, another interesting question is what are the functional differences that differentially activate GCN2 and PERK. The authors show that ER stress is similar between these two types of cell lines, but is this differential response dictated in part through altered metabolic signaling through reduced amino acids and/or mitochondrial imbalances, both of which can activate GCN2. Taking a step backward by showing that Mal3-101 differentially targets some aspect of metabolism to increase dependence on GCN2 in resistant cells would be quite interesting and significantly boost the impact of this work.

3. Alternatively, the authors could take a more therapeutic approach by incorporating the use of selective GCN2 or PERK inhibitors in this work, showing that combinations of GCN2 inhibitors and Mal3-101 increase apoptosis of resistant cells while more selective PERK inhibitors block apoptosis in the sensitive cells.

Other Comments.

1. I realize that the Mal3-101 compound has been reported previously, but considering the way it is used, it would be good to have a statement up front in the text about its selectivity and known engagement of different HSP70s. Which ones it hits, EC50, etc. For example, the potential for this compound to target BiP?. What about mitochondrial HSP70? It would be good to have a better description of the compound up front in this manuscript to help better understand its effects on different cancer cell lines. This is addressed in the results and discussion, but it would be good to include this more clearly in the intro.

2. It would be good to directly confirm the differential activation of PERK in GCN2 in sensitive and resistant cells more directly. For example, a phosphor-PERK and phosphor-GCN2 assay to show increased activity of these two eIF2a kinases.

I strongly encourage the authors to continue pursuing the interesting questions related to differential signaling through ISR kinases using the system they have established. Above, I provide some suggestions as to how to bring a bit of closure to this initial manuscript, which I imagine will represent the first of many pursuing this question.

*Reviewer #2:*

This article explores the response of cultured breast cancer cell lines to an inhibitor of heat shock protein 70 (Hsp70). The idea is to understand how different cells might be more/less sensitive to the same proteostatic stress. The concept of drug resistance in proteostasis is interesting and has only been studied extensively for inhibitors of Hsp90 and the 26S proteasome, so a significant amount is still unclear.

Here, the authors use the Hsp70 inhibitor, MAL3-101, and some observations from a recent study in rhabdomyosarcoma, to treat a panel of 14 breast cancer cells, ultimately placing them into categories based on their relative sensitivity (see table 1). To understand the origins of the sensitivity, they then use a hypothesis-driven approach to explore possible correlations with relative Hsp70 expression levels, ubiquitinated protein accumulation, autophagic flux and UPR activation. From this analysis, the only major correlation seemed to be with compensatory activation of autophagy through GCN2, which gave relative resistance to the inhibitor. While the mechanism of this difference remains unclear, this finding is potentially interesting for translation of the compounds (e.g. for patient stratification). More broadly, the results suggest that different cell types have inherently individualized tendencies to activate components of the proteostasis network (i.e. different arms of the UPR, autophagy vs. UPS). While perhaps not a surprising result when considered abstractly, the approach taken here, use of a chemical inhibitor to probe relative sensitivity, is powerful because it can be rapidly applied across many cell lines.

While the work is interesting, there does seem to be a missed opportunity because, in the end, one still wonders about the mechanism of the differential resistance. One idea to consider: there are publicly available transcriptional profiles of many of these breast cancer cell lines, so a clustering analysis might reveal the patterns of transcription that define the sensitivity groups, shedding light on potential mechanisms in an unbiased way. Does sensitivity correlate to known driver oncogenes? Drug efflux protein expression? Amino acid transporters? Metabolism? Translation control machinery? More colloquially – the classical breast cancer categorization subtype (e.g. TNBC, luminal, HER2) do not correlate with Hsp70 inhibitor sensitivity, so what does at a molecular level? Similarly, the authors also systematically examined resistance to the Hsp90 inhibitor, 17-AAG, so a parallel analysis of transcriptional profiles could provide further insight into the similarities and differences.

A second question is about whether GCN2 activation is necessary/sufficient for Hsp70 inhibitor resistance. Does over-expression of GCN2 drive resistance in an otherwise sensitive cell type? How about amino acid starvation? mTor inhibitor? It seems likely that the story could be more complex, but a specific test would be informative either way.

*Reviewer #3:*

The manuscript "Unique integrated stress response sensors regulate cancer cell susceptibility when Hsp70 activity is compromised" by Sannino and colleagues describes that autophagy confers resistance against HSP70 inhibition in cancer treatment. They grouped breast cancer cell lines according to their resistance to the HSP70 inhibitor MAL3-101 and showed that resistant cell lines exhibited higher levels of autophagy induction after MAL3-101 treatment when compared to the sensitive cells. Autophagy inhibition resensitized the resistant cells to MAL3-101 treatment, indicating autophagy as the underlying resistance mechanism. In the second part of the paper, the authors show that the unfolded protein response and the integrated stress response (ISR) were activated upon HSP70 inhibition and examined the role of the different eIF2a kinases activating the ISR.

Overall, the paper is well written and addresses an important topic. Chaperone inhibition has long been a target in cancer therapy and the connections to autophagy and cellular stress responses remain unclear. The study adds interesting new insight that will be of interest across the proteostasis, autophagy and cancer fields. The authors convincingly show different behaviour of tumor subgroups when treated with an HSP70 inhibitor and offer options to overcome resistance formation. However, some mechanistic details on the links with the ISR and autophagy flux reporters would need to be characterised better to make this paper stronger.

1) The mechanisms leading to eIF2a phosphorylation and whether the resulting phenotypes are indeed driven by the ISR remain unclear. A) The arguments towards omitting HRI from the analyses are not convincing. A large number of studies have shown that HRI activation is not exclusive to mitochondrial dysfunction (see e.g. PMID: 10454533). HRI also needs to be evaluated along with the other eIF2a kinases. B) Why is the ISR activated both in sensitive and resistant cells while the cell fate is different? This raises the possibility that the observed effects are driven by another pathway (several pathways have been shown to induce CHOP, ATF4, autophagy). Further tests using ISRIB or phosphorylation incompetent versions of eIF2a will be required to clarify the requirement of ISR activation. C) The 3 eIF2a kinases assessed show varying results across the different cell groups, treatments and outcomes. Additional clarification and interpretation of these seemingly contradictory results would be helpful. D) Considering the above points, there is a chance that the eIF2a knockdowns were not sufficient to prevent eIF2a phosphorylation. eIF2a phosphorylation Western blots after knockdown of the different kinases and treatment with positive controls (tunicamycin for PERK…) would clarify this issue.

2) Autophagy flux reporters, such as RFP-GFP-LC3 used by the authors, have become standard in the field. These are much more reliable than assessment of LC3 lipidation that can be hard to interpret or autophagy protein levels. It is a pity the authors did not assess flux more consistently. It would be crucial using the flux reporter to assess A) at least one more additional sensitive and resistant cell line for the effects of MAL3-101 (as in Figure S5), B) the proposed autophagy induction via the ISR. These experiments will much strengthen the proposed role of autophagy.

---

## [Author Response]

Reviewer #1:[…] Overall, the authors have clearly performed numerous experiments to define the protein homeostasis remodeling associated with Mal3-101 sensitivity in breast cancer cell lines. Further, their concluding results that sensitive and resistant cells are differentially dependent on GCN2 and PERK for dictating survival or apoptosis, respectively are interesting. However, the idea that GCN2 and PERK (or other ISR kinases) can initiate distinct signaling in response to different types of insults is not exactly a new idea in the field, although admittedly, this work does organize this idea in a defined way and adds new results that can continue to inform that area of study. The key question is how does ISR signaling through PERK and GCN2 integrate with other functional inputs to dictate distinct cellular responses. This is likely achieved through mechanism including the posttranslational modification of different downstream transcriptional regulators, integration of ISR signaling with other stress-responsive signaling pathways, or through heterooligomerization of ISR transcription factors with other inputs (e.g., heterooligomerization of ATF4 with other b-ZIP transcription factors). The authors work uniquely positions them to address these types of questions, but ultimately, they are not pursued in this manuscript. In the current form, the manuscript sets up the next series of very interesting studies to get the bottom of this type of differential regulation. To be fair, it is not reasonable to expect the entire underlying mechanism highlighted by this work to be defined in a single paper, but I do think it is important to address some aspects of the differential sensitivity from either a biological or therapeutic perspective.Below I include a few suggested experiments to bring some more closure to the manuscript. All are not necessary, especially considering the amount of work already included, but one of these types of experiments would significantly improve the impact of the work. There are, of course, other potential experiments too that would be suitable to bring closure to this story, but I think something similar to the below is required.1. One interesting question emerging from this study is: 'what are the other inputs for resistant and sensitive cells that differentially alter ISR signaling outputs in these two models?'. This could potentially be addressed by performing RNAseq of resistant or sensitive cells treated with Mal3-101 and depleted of either Gcn2 or Perk. This would provide a better framework to interpret the differential ISR signaling between these two types of cell lines and likely reveal other signaling mechanism that serve as modulators of ISR signaling.

The reviewer makes an excellent suggestion, and to address this point as well as the question raised in point 2, below, we analyzed the transcriptional profiles of a MAL3-101 sensitive and resistant cell line (MDA MB 231 and MDA MB 453) in the presence of 12 µM MAL3-101 for 6 hr (when robust transcriptional responses for the autophagy pathway were noted by qPCR) or in an equal volume of DMSO. Each experiment was performed in triplicate. We next performed total RNA extraction and performed a quality control analysis. RNA sequencing was performed by Novogene and analyzed by three new co-authors on the manuscript (Drs. Megan Yates, Steffi Oesterreich, and Adrian Lee). These colleagues are experts in the analysis of cancer cell genomics and transcriptomics. As shown in a new figure (Figure 13), we discovered that protein folding and ER stress response pathways are upregulated in sensitive cells (MDA MB 231), while select metabolic pathway components are upregulated in resistant cells (MDA MD 453) after MAL3-101 treatment. The RNAseq analysis shown in Figure 13 also confirm that Hsp70 inhibition causes unfolded proteins to accumulate and the consequent activation of the unfolded protein response (UPR). These results confirm other data in the manuscript that UPR induction, and in particular activation of the PERK branch of this pathway, induces MAL3-101-dependent cell death in sensitive cells (MDA MB 231). We also wish to note that we refrained from performing these experiments under specific knock-down conditions since secondary effects are likely, particularly in the sensitive cells, which initiate a global apoptotic program in the presence of MAL3-101 in the same time frame.

2. Similar to #1, another interesting question is what are the functional differences that differentially activate GCN2 and PERK. The authors show that ER stress is similar between these two types of cell lines, but is this differential response dictated in part through altered metabolic signaling through reduced amino acids and/or mitochondrial imbalances, both of which can activate GCN2. Taking a step backward by showing that Mal3-101 differentially targets some aspect of metabolism to increase dependence on GCN2 in resistant cells would be quite interesting and significantly boost the impact of this work.

Please see the comments to point 1, above. Please also note that hierarchical clustering the samples based upon the top 35 differential expressed genes in these cells reveals that MAL3-101 treated sensitive (MDA MB 231) cells cluster with untreated resistant MDA MB 453 cells, while no major variations were detected in the transcriptional profile of MAL3-101 resistant cells (MDA MB 453) in presence or absence of the drug (Figure 13C and13D). These findings suggest that Hsp70 inhibition perturbs translational and/or metabolic pathways rather than transcription in MAL3-101 resistant cells. This hypothesis will be pursued in full in future studies. In addition, we note that an autophagy related gene, MAP1LC3B, was highly induced upon MAL3-101 treatment in MDA MB 453 resistant cells. These data further support our conclusions that autophagy is induced in resistant cells upon Hsp70 inhibition.

3. Alternatively, the authors could take a more therapeutic approach by incorporating the use of selective GCN2 or PERK inhibitors in this work, showing that combinations of GCN2 inhibitors and Mal3-101 increase apoptosis of resistant cells while more selective PERK inhibitors block apoptosis in the sensitive cells.

This is an excellent point, one that we had considered as well. Therefore, to address this comment, we treated sensitive and resistance cells with MAL3-101 in the presence or absence of the PERK inhibitor, GSK-2606414 (“GSK”). Cells were pre-treated for 2 hours with this PERK inhibitor (or a vehicle control) to specifically block this ISR sensor, and then MAL3-101 or DMSO was added for an additional 6 hours. The magnitude of apoptotic induction under each of these conditions is reported in a revised figure, Figure 10D, and quantified in Figure 10—figure supplement 1D. As predicted by our PERK silencing experiments and consistent with one of our major conclusions, there was reduced accumulation of cleaved caspase-3, cleaved caspase-8, and CHOP in sensitive cells when PERK was inhibited upon MAL3-101 addition. Also as anticipated, there was apoptotic induction with MAL3-101 resistant cells when GSK was added in conjunction with the Hsp70 inhibitor. Therefore, while an ISR-based response is induced in both sensitive and resistant cells, PERK contributes to MAL3-101-mediated apoptosis in only the MAL3-101 sensitive cells.

Other Comments.1. I realize that the Mal3-101 compound has been reported previously, but considering the way it is used, it would be good to have a statement up front in the text about its selectivity and known engagement of different HSP70s. Which ones it hits, EC50, etc. For example, the potential for this compound to target BiP?. What about mitochondrial HSP70? It would be good to have a better description of the compound up front in this manuscript to help better understand its effects on different cancer cell lines. This is addressed in the results and discussion, but it would be good to include this more clearly in the intro.

We have included more information on the isolation, characterization, and uses of the MAL3-101 compound in the Introduction section, as suggested (see page 7, lines 123-127).

2. It would be good to directly confirm the differential activation of PERK in GCN2 in sensitive and resistant cells more directly. For example, a phosphor-PERK and phosphor-GCN2 assay to show increased activity of these two eIF2a kinases.

To detect PERK activation (p-PERK), we monitored the shift in PERK migration by immunoblot after SDS-PAGE after sensitive and resistant cells were treated with a known UPR inducer, tunicamycin, MAL3-101, or the vehicle control. As shown in a new figure (Figure 10C), PERK mobility was reduced to a similar extent in MAL3-101 sensitive and resistant cells when Hsp70 was inhibited or when the cells were treated with tunicamycin. These data confirm that PERK is activated in both MAL3-101 sensitive and resistant cells upon Hsp70 inhibition.

To monitor GCN2 activation, the level of phosphorylated GCN2 at Thr-899 was investigated by using a pT899 specific antibody (anti-GCN2 pT899 antibody ab-75836, Abcam; see Figure 11—figure supplement 7A) (Romano et al., 1998). Interestingly, pT899 was apparent even before MAL3-101 was added in both sensitive and resistant cells, suggesting a requirement for activated GCN2 in the survival and perhaps immortalization of these breast cancer cells. This was therefore independent of whether Hsp70 was inhibited or not. Consistent with this hypothesis, no further change in pT899 GCN2 was detected when MAL3101 was added to either cell line (Figure 11—figure supplement 1A). Nevertheless, it is possible that other sites are being phosphorylated when Hsp70 is inhibited. In fact, there was a hint of this occurring in a previous study (Harding et al., 2019) in which the role of the ribosomal Pstalk on GCN2 activation was investigated. A discussion of these points and the relevant additional reference have been added to both the Results section of the manuscript (see page 25 in the Results section).

I strongly encourage the authors to continue pursuing the interesting questions related to differential signaling through ISR kinases using the system they have established. Above, I provide some suggestions as to how to bring a bit of closure to this initial manuscript, which I imagine will represent the first of many pursuing this question.Reviewer #2:This article explores the response of cultured breast cancer cell lines to an inhibitor of heat shock protein 70 (Hsp70). The idea is to understand how different cells might be more/less sensitive to the same proteostatic stress. The concept of drug resistance in proteostasis is interesting and has only been studied extensively for inhibitors of Hsp90 and the 26S proteasome, so a significant amount is still unclear.Here, the authors use the Hsp70 inhibitor, MAL3-101, and some observations from a recent study in rhabdomyosarcoma, to treat a panel of 14 breast cancer cells, ultimately placing them into categories based on their relative sensitivity (see table 1). To understand the origins of the sensitivity, they then use a hypothesis-driven approach to explore possible correlations with relative Hsp70 expression levels, ubiquitinated protein accumulation, autophagic flux and UPR activation. From this analysis, the only major correlation seemed to be with compensatory activation of autophagy through GCN2, which gave relative resistance to the inhibitor. While the mechanism of this difference remains unclear, this finding is potentially interesting for translation of the compounds (e.g. for patient stratification). More broadly, the results suggest that different cell types have inherently individualized tendencies to activate components of the proteostasis network (i.e. different arms of the UPR, autophagy vs. UPS). While perhaps not a surprising result when considered abstractly, the approach taken here, use of a chemical inhibitor to probe relative sensitivity, is powerful because it can be rapidly applied across many cell lines.While the work is interesting, there does seem to be a missed opportunity because, in the end, one still wonders about the mechanism of the differential resistance. One idea to consider: there are publicly available transcriptional profiles of many of these breast cancer cell lines, so a clustering analysis might reveal the patterns of transcription that define the sensitivity groups, shedding light on potential mechanisms in an unbiased way. Does sensitivity correlate to known driver oncogenes? Drug efflux protein expression? Amino acid transporters? Metabolism? Translation control machinery? More colloquially – the classical breast cancer categorization subtype (e.g. TNBC, luminal, HER2) do not correlate with Hsp70 inhibitor sensitivity, so what does at a molecular level? Similarly, the authors also systematically examined resistance to the Hsp90 inhibitor, 17-AAG, so a parallel analysis of transcriptional profiles could provide further insight into the similarities and differences.

As suggested by the reviewer, we analyzed available databases containing transcriptional profiles and proteomic data for the 14 breast cancer lines used in this study in order to identify pathways that might differentiate MAL3-101 sensitive and resistant breast cancer cells. As displayed in a new figure (Figure 13A and 13B), MAL3-101 sensitivity did not correlate with changes in breast cancer cell transcriptional profiles, suggesting that MAL3-101 sensitivity is related to translational variations rather than changes in gene expression. In line with this hypothesis, the analysis of publicly available RPPA datasets revealed that pT172 AMPK and p70-S6K-pT389 were present to a higher extent in MAL3-101 resistant cells (MDA MB 453, MDA MB 361 and HCC38) compared to the sensitive cell lines (Figure 13B). pT172 AMPK and p70-S6K-pT389 orchestrate the modulation of cellular metabolism and protein translation when cellular energy varies, suggesting that MAL3-101 resistant cells are more likely to modulate their metabolic state upon proteostasis stress. This hypothesis will be investigated in full in future studies. Interestingly, a previous study indicated that pT172 AMPK and p70-S6K-pT389 upregulation was also detected together with autophagy induction in HEK293 cells deprived of glucose (Kim et al., 2011). These findings provide further support for our conclusion that autophagy is a key compensatory pathway in resistant cells treated with MAL3-101. A discussion of these points – as well as the possibility that MAL3-101 alters AMPK activity and thus cellular metabolism – and the relevant references have been added to the “Induction of the ER stress response and protein folding pathway genes correlate with Hsp70 inhibition in sensitive cells” paragraph.

A second question is about whether GCN2 activation is necessary/sufficient for Hsp70 inhibitor resistance. Does over-expression of GCN2 drive resistance in an otherwise sensitive cell type? How about amino acid starvation? mTor inhibitor? It seems likely that the story could be more complex, but a specific test would be informative either way.

The reviewer raises two excellent points.

First, to investigate whether GCN2 was necessary and sufficient to confer MAL3-101 sensitivity, we overexpressed Flag-tagged GCN2 in sensitive and resistant cells, as suggested. Unfortunately, the overexpression of GCN2 alone caused the accumulation cleaved caspase-8 compared to vector-control transfected cells (Author response image 1).

This result suggests that GCN2 overexpression per se is sufficient to trigger an apoptotic response. In theory, this problem might be rectified by creating stable, tunable GCN2 over-expression lines, a feat that will require a significant amount of work and might still indicate that overexpression of GCN2 – to any level – is toxic. Indeed, we were surprised that we were unable to find any papers in the literature in which GCN2 overexpression studies were performed, potentially supporting the phenomenon we observed.

**Author response image 1. sa2fig1:** GCN2 overexpression induces apoptosis. Accumulation of cleaved caspase -8, which is an indicator of apoptosis induction, was monitored in empty vector (CTRL) and Flag -GCN2 transfected MAL3-101 sensitive and resistant cells.

Second, to investigate the role of mTOR in MAL3-101 sensitivity, we first tested MAL3-101 sensitive and resistant cell viability in presence or absence of the mTOR inhibitor, everolimus. As shown in a new figure (Figure 6C), only subtle differences were detected in the sensitivity to this inhibitor in the two representative cell lines (IC_50_ 23.0 µM for MDA MB 231 cells and 18.7 µM for MDA MB 453 cells). However, when the sensitive line (MDA MB 231) was treated with increasing doses of MAL3-101 in the presence of a submaximal concentration of everolimus (2 µM), we discovered that autophagy induction via this mTOR inhibitor modestly decreased the sensitivity of MDA MB 231 cells to MAL3-101 (Figure 6D and Table 2, IC_50s_ MAL3-101 alone 3.3 µM, MAL3-10 + everolimus 5.1 µM). In contrast addition of sub-critical doses of everolimus in the presence of MAL3-101 had no effect on cell viability in the resistant (MDA MB 453) cell line. These data confirm another one of our conclusions: Autophagy provides a compensatory pro-survival mechanism when Hsp70 activity is inhibited. These data also highlight the importance of monitoring protein degradation efficiency when analyzing chaperone-based inhibitors.

Reviewer #3:The manuscript "Unique integrated stress response sensors regulate cancer cell susceptibility when Hsp70 activity is compromised" by Sannino and colleagues describes that autophagy confers resistance against HSP70 inhibition in cancer treatment. They grouped breast cancer cell lines according to their resistance to the HSP70 inhibitor MAL3-101 and showed that resistant cell lines exhibited higher levels of autophagy induction after MAL3-101 treatment when compared to the sensitive cells. Autophagy inhibition resensitized the resistant cells to MAL3-101 treatment, indicating autophagy as the underlying resistance mechanism. In the second part of the paper, the authors show that the unfolded protein response and the integrated stress response (ISR) were activated upon HSP70 inhibition and examined the role of the different eIF2a kinases activating the ISR.Overall, the paper is well written and addresses an important topic. Chaperone inhibition has long been a target in cancer therapy and the connections to autophagy and cellular stress responses remain unclear. The study adds interesting new insight that will be of interest across the proteostasis, autophagy and cancer fields. The authors convincingly show different behaviour of tumor subgroups when treated with an HSP70 inhibitor and offer options to overcome resistance formation. However, some mechanistic details on the links with the ISR and autophagy flux reporters would need to be characterised better to make this paper stronger.1) The mechanisms leading to eIF2a phosphorylation and whether the resulting phenotypes are indeed driven by the ISR remain unclear.A) The arguments towards omitting HRI from the analyses are not convincing. A large number of studies have shown that HRI activation is not exclusive to mitochondrial dysfunction (see e.g. PMID: 10454533). HRI also needs to be evaluated along with the other eIF2a kinases.

The reviewer makes an excellent suggestion, and we apologize for not appreciating the potential role of HRI in this and other functions. Thus, in the revised manuscript, we present new data ont the role of HRI in the MAL3-101-dependent induction of apoptosis (Figure 9). Specifically, we found that HRI knockdown had no effect on the viability of the resistant (MDA MB 453) cells. On the contrary, accumulation of cleaved caspase-8 and CHOP, but not cleaved caspase-3, was detected in MDA MB 231 sensitive cells when MAL3101 was added to HRI-silenced cells. While the data did not reach statistical significance, these results suggest that HRI plays a minor role in the MAL3-101-initiated cellular response in drug-sensitive cells (Figure 9A and 9C). A discussion of this model has been added to the revised manuscript.

B) Why is the ISR activated both in sensitive and resistant cells while the cell fate is different? This raises the possibility that the observed effects are driven by another pathway (several pathways have been shown to induce CHOP, ATF4, autophagy). Further tests using ISRIB or phosphorylation incompetent versions of eIF2a will be required to clarify the requirement of ISR activation.

We thank the reviewer for this comment, and to address their excellent suggestion a new figure has been included the revised manuscript. Specifically, we measured the MAL3101-dependent apoptotic response in presence or absence of the ISR pathway inhibitor, ISRIB, as suggested (Figure 11—figure supplement 1). In line with our data on the effects of PERK silencing and chemical inhibition of this ISR transducer (Figure 10 and Figure 10—figure supplement 1), ISRIB treatment decreased the accumulation of cleaved caspase-3, cleaved caspase-8, and CHOP in MDA MB 231 sensitive cells when MAL3-101 was added. These data confirm that the ISR response does indeed initiate the apoptotic pathway in MAL3-101 sensitive cells when Hsp70 is inhibited.

In contrast to the measurable protective effect of ISRIB in the sensitive cells, only a minor change in one of the examined apoptotic markers, cleaved caspase-8, was apparent when Hsp70 was inhibited in the resistant (MDA MB 453) cell line upon ISRIB addition. These data suggest that inhibition of the ISR-eIF2α signaling pathway is insufficient to induce resistant cell apoptosis. Because this effect was subtle compared to the apoptotic effect when GCN2 expression was silenced and MAL3-101 was added, we hypothesize that another ISR sensor helps dictate MAL3-101-dependent apoptotic cell death in resistant cells in the presence of the drug. This hypothesis is outlined in the revised manuscript in conjunction with these newly acquired data (see Figure 11—figure supplement 1).

C) The 3 eIF2a kinases assessed show varying results across the different cell groups, treatments and outcomes. Additional clarification and interpretation of these seemingly contradictory results would be helpful. D) Considering the above points, there is a chance that the eIF2a knockdowns were not sufficient to prevent eIF2a phosphorylation. eIF2a phosphorylation Western blots after knockdown of the different kinases and treatment with positive controls (tunicamycin for PERK…) would clarify this issue.

As suggested by the reviewer, we addressed the effect of PERK silencing when a known UPR stressor (in this case, tunicamycin) was added to MAL3-101 sensitive and resistant cells. As quantified in a new figure, Figure 10—figure supplement 1B, PERK knockdown decreased the amount of p-eIF2α and CHOP in both sensitive and resistant cells treated with 10 µg/mL tunicamycin for 3 hours. Thus, the level of PERK knockdown achieved in our experiments is also sufficient to reduce PERK-p-eIF2α induction in sensitive and resistant cells treated with MAL3-101, which similarly initiates the ISR.

2) Autophagy flux reporters, such as RFP-GFP-LC3 used by the authors, have become standard in the field. These are much more reliable than assessment of LC3 lipidation that can be hard to interpret or autophagy protein levels. It is a pity the authors did not assess flux more consistently. It would be crucial using the flux reporter to assess A) at least one more additional sensitive and resistant cell line for the effects of MAL3-101 (as in Figure S5), B) the proposed autophagy induction via the ISR. These experiments will much strengthen the proposed role of autophagy.

We thank the reviewer for appreciating our development of the stable cell line in which autophagy is best measured. In fact, the construction, isolation, and verification of this line required ~ 6 months. Nevertheless, we note that a less-than-perfect alternate method to analyze autophagy induction, i.e., LC3 immunoblots, do show a marked difference in autophagy between two sensitive and two resistant lines, including the lines in which RFPGFP-LC3 was also used (Figure 2D, 4C and 4D). When taken together, these data indicate that the outcome of measuring LC3 by immunoblot mirrors the effects in the resistant and sensitive lines containing the stably integrated version of RFP-GFP-LC3.